# Changes of Target Essential Trace Elements in Multiple Sclerosis: A Systematic Review and Meta-Analysis

**DOI:** 10.3390/biomedicines12071589

**Published:** 2024-07-17

**Authors:** Aleksandar Stojsavljević, Jovana Jagodić, Tatjana Perović, Dragan Manojlović, Slađan Pavlović

**Affiliations:** 1Innovation Center, Faculty of Chemistry, University of Belgrade, Studentski Trg 12-16, 11000 Belgrade, Serbia; 2Faculty of Chemistry, University of Belgrade, 11000 Belgrade, Serbia; jovanaj@chem.bg.ac.rs (J.J.); manojlo@chem.bg.ac.rs (D.M.); 3Psychiatric Hospital, University Medical Center Zvezdara, 11000 Belgrade, Serbia; tzogovic@gmail.com; 4Serbian RE&CBT Centre, 11000 Belgrade, Serbia; 5Institute for Biological Research “Siniša Stanković”-National Institute of the Republic of Serbia, University of Belgrade, 11108 Belgrade, Serbia; sladjan@ibiss.bg.ac.rs

**Keywords:** multiple sclerosis (MS), essential trace elements, deficiency, supplementation, monitoring

## Abstract

(1) Background: Multiple sclerosis (MS) is a chronic, complex, and demyelinating disease closely associated with altered levels of trace elements. Although the first studies into the role of trace elements in MS were published in the 1970s, for five decades it has remained unknown whether trace elements can be part of this heterogeneous neurological disease. (2) Materials and methods: To drive toward at a potential solution, we conducted a systematic review and meta-analysis to elucidate whether there were differences in circulating levels of neurologically important essential trace elements (Zn, Fe, Co, Cu, Mn, and Se) between MS cases and controls. (3) Results: This study revealed significantly lower serum/plasma Zn and Fe levels and higher Cu levels in MS-affected individuals compared to controls. At the same time, no significant differences were found between the MS cases and controls regarding their serum/plasma levels of Co, Mn, or Se. Thus, the loss of Fe and Zn should be considered in supplementation/nutrition strategies for MS patients. On the other hand, since high serum Cu levels indicate a burden on the bloodstreams of MS patients, Cu should be excluded from mineral supplement strategies. Furthermore, all three trace elements (Fe, Zn, and Cu) should be considered from an etiological point of view, and, most importantly, their levels in the bloodstreams of MS patients should be monitored. (4) Conclusions: This study highlights the way for personalized and targeted strategies in the management of MS.

## 1. Multiple Sclerosis and Trace Elements

Multiple sclerosis (MS) is a progressive, chronic illness of the central nervous system (CNS) that impacts an estimated 2.8 million people across the globe [1,2]. This intricate disease manifests in an array of physical manners, ranging from sensory, visual, and motor difficulties to the development of total disability [3,4]. It is distinguished by gradual neurodegeneration of the brain and spinal cord that appears in the form of sclerotic plaques signifying demyelination of the white and gray matter [5]. Weakness, fatigue, bladder issues, muscle spasms, dizziness, sensory issues like double vision, visual abnormalities, and/or discomfort are common symptoms reported by people with MS [6]. MS is commonly showcased by worsening of symptoms preceding months-long remissions with practically complete symptom alleviation. Over time, symptoms typically worsen and remission periods become less complete [7]. MS is more prevalent in females than males and is frequently identified between the ages of the early 20s and the 40s [8,9]. Only about 3–10% of all MS cases are diagnosed in childhood years [8].

The diagnostic criteria for MS, which combine clinical, imaging and laboratory findings, have evolved over time. In 1996, the Advisory Committee on Clinical Trials in Multiple Sclerosis Society (NMSS) Advisory Committee on Clinical Trials in Multiple Sclerosis defined the clinical subtypes of MS to describe the different clinical courses of MS. There was a need for clarification in order to improve homogeneity in the clinical field [10]. Taking into account the clinical presentation, Schaeffer et al. [11] distinguished four types of MS: relapsing–remitting MS (RRMS), secondary progressive MS (SPMS), primary progressive MS (PPMS), and progressive relapsing MS (PRMS). RRMS is the most commonly diagnosed form of MS, occurring in around 85% of cases [12], with relapses occurring one or more months or even several years apart. Patients who relapse either recover completely without persistent neurological problems or recover partially. SPMS is a progressive form of RRMS characterized by impaired general functioning. Approximately 50% of people with RRMS develop SPMS in the 15–20 years following their initial diagnosis [12]. The hallmarks of MS, such as relapses, are not recognizable in PPMS; instead, there is a progressive loss of neurological function over time. Only 10% of MS patients are diagnosed with PPMS. PRMS, which is considered the rarest form of MS, is characterized by possible abrupt relapses during the course of the disease, in addition to a progressive worsening of symptoms after their initial onset [12,13]. The descriptions of the clinical course from 1996 were quickly adopted in clinical practice and used in the approval criteria for almost all subsequent clinical MS studies [10].

In 2012, following a re-examination of MS phenotypes, research into clinical, imaging, and biomarker advances by working groups and through literature searches, the clinical course descriptions were reviewed, and it was determined whether there were some potential advances and new findings to recommend changes [10]. In short, based on experience, some changes were introduced: the assessment of disease activity as defined by clinical assessment of relapse occurrence or lesion activity detected by CNS imaging, the determination of whether disability progression has occurred in a given period of time, the categorization of PP patients with disease activity as PRMS, and the establishment of PPMS as part of the spectrum of progressive disease. In addition, new terms were introduced: clinically isolated syndrome (CIS), in which features of inflammatory demyelination that could be MS but do not yet meet the criteria for dissemination), which was added to the spectrum of MS phenotypes; and radiologically isolated syndrome (RIS, in which imaging findings suggest inflammatory demyelination in the absence of clinical signs or symptoms) which was not considered a separate MS phenotype [10].

One of the most commonly used sets of criteria for the diagnosis of MS is the 2010 McDonald criteria. The 2010 McDonald criteria were modified based on recommendations on the diagnostic process for MS, resulting in the 2017 McDonald criteria [14]. The specific updates of the 2010 McDonald criteria to the 2017 McDonald criteria include the distinction between MS and other conditions with potentially overlapping clinical and imaging features, such as Neuromyelitis Optica Spectrum Disorders (NMOSDs), challenges in diagnosing individuals with presentations that do not represent a typical clinically isolated syndrome, the frequency and consequences of misdiagnosis, and CSF and other paraclinical tests that could be used to diagnose MS [14].

The timely and accurate diagnosis of MS remains a major challenge for clinicians. Despite all the improvements and the use of new technologies, we have not yet reached the point where we can be sure of making the correct diagnosis in time, which is crucial for further treatment. The early diagnostic criteria for MS were primarily based on clinical findings. Later criteria incorporated imaging and other paraclinical markers in response to technological advances and new data. Since there is no particular test for MS, diagnosis is typically difficult to obtain, especially in the early stages of the illness [15]. The verification of MS is made by adequately disseminating information and ruling out other conditions with similar symptoms [16]. For this purpose, the McDonald criteria are employed and serve as a set of recommendations for physicians [17]. These criteria are composed of multiple tests, i.e., magnetic resonance imaging of the brain and spinal cord, various blood tests, lumbar puncture, and nerve activity tests, all of which are integrated to provide a suitable diagnosis [18]. There is no proper cure for MS. The adequate management of symptoms and flare-ups by employing various medications is currently the only possible way of handling this disease [19].

Despite tremendous efforts over the past few decades to shed more light on this widespread CNS disease, the etiology of MS remains unclear [20]. According to some studies, trace metals could potentially contribute to the onset of MS. Furthermore, it has been shown that working in mechanical, steel, and leather industries was linked with a greater chance of acquiring this disease [21,22]. Similarly, places with excessive toxic metal pollution are linked with an increased prevalence of MS in the resident populations [21]. Toxic trace elements can cause a negative health impact by generating free radical species, suppressing enzymatic activity, and influencing the uptake of essential trace metals [23]. On the other hand, Mezzaroba et al. (2019) [24] reported that changes in the levels of essential trace metals, such as Zn, Cu, and Mn, have been linked with MS. These trace elements, along with Fe, Se, and Co, are required for proper synaptic transmission, myelination of neurons, neurotransmitter activity, and the CNS’s signal transduction pathways [25,26,27]. Furthermore, these metals have roles in CNS development, antioxidative defense, and protection from cellular damage. The regulation of these metals’ concentrations in the CNS is controlled by complex brain barrier systems, such as the blood-brain barrier (BBB), blood–cerebrospinal fluid (CSF) barrier, choroidal blood–cerebrospinal fluid barrier, and CSF–brain barrier [28]. When the delicate balance of these metals is disrupted, morphofunctional alterations in the CNS can occur, leading to the emergence of a variety of health impairments. Specifically, Zn, as an essential trace metal and one of the most abundant elements in the CNS, has a pivotal role in neurogenesis and neurotransmitter activity [26]. Copper is present throughout the brain, but is particularly abundant in the basal ganglia, hippocampus, cerebellum, and numerous synaptic membranes. This trace metal is required for the proper synaptic transmission and myelination of neurons [25]. Impaired neuronal physiology and cognition have been tied to the deregulation of Mn levels [29]. Iron serves functionally in myelin and neurotransmitter production, mitochondrial respiration, and oxygen transport. Deregulated Fe levels have been associated with neurotoxicity and MS pathophysiology [30,31]. Aside from its established role in the form of cobalamin, Co, as its dust, can cause neuropathy in occupationally exposed workers [32]. Selenium plays a crucial part in the CNS’s signal transduction pathways, by influencing cognition and motor functions [27].

The particular mechanisms of these elements’ neurotoxicity are yet to be fully understood, although experimental and epidemiological evidence of their roles in MS onset has grown over time. In this regard, we conducted a systematic review and meta-analysis to elucidate whether there were differences in circulating levels of neurologically important essential trace elements (Zn, Fe, Co, Cu, Mn, and Se) between MS cases and controls.

## 2. Materials and Methods

The baseline for this systematic review and meta-analysis is the “Preferred Reporting Items for Systematic Reviews and Meta-Analyzes: the PRISMA Statement” [33], which was originally proposed in 2009. In this meta-analysis, we used the updated PRISMA 2020 statement, which includes new reporting guidelines that reflect methodological advances in the identification, selection, appraisal, and synthesis of studies [34] (Appendix A).

### 2.1. Information Sources

Four databases were initially used to search for articles that could be incorporated into this study: PubMed, SCOPUS, ScienceDirect, and Google Scholar. These databases were found by us to partially overlap, and so we focused solely on the two most representative databases (those that produced the greatest number of results), PubMed and ScienceDirect.

### 2.2. Search Strategy

We conducted our literature search to encompass years from 1976 to 2024. Our first aim was to identify research articles with studies that investigated the concentrations of Zn, Fe, Co, Cu, Mn, and Se in the serum/plasma of patients with MS (cases) and in healthy individuals (controls). We conducted a systematic search using the following keywords: “multiple sclerosis” AND “serum” AND “plasma” AND “trace elements” AND “zinc” AND “iron” AND “cobalt” AND “copper” AND “manganese” AND “selenium”.

In our search, we found a total of 5051 MS-related author keywords, with 672 author keywords connecting MS and trace elements and, finally, 27 keywords that were the most frequently used. The comprehensive network visualization and the relationships between the observed keywords can be found in Figure 1A–C (created with the software VOSviewer 1.6.19, Copyright (c) 2009–2023 Nees Jan van Eck and Ludo Waltman Center for Science and Technology Studies of Leiden University, Leiden, The Netherlands).

After the search, we carefully checked the reference lists of the research articles found. Our inclusion criteria were original case–control, cohort or cross-sectional studies in which the levels of the trace elements of interest (Zn, Fe, Co, Cu, Mn and/or Se) in the specified clinical matrices were reported by both MS and control subjects. Exclusion criteria were studies in which the MS diagnosis was not confirmed, studies with cases and controls not from the same geographical area (place of residence), studies with age- and gender-atypical cases and/or controls, studies reporting additional pathologies besides MS, non-English-language studies, studies with insufficient numerical data, and studies with extremely analytically abnormal levels of the quantified trace elements. We excluded studies that reported only the mean without SD or SE and those studies that showed only graphs without numerical values. We included studies that included results where the mean ± standard deviation (SD) or standard error (SE, from which we could calculate the SD) was reported. We also included studies with other numerical data from which we could calculate the SD (see Section 2.3, below). We used tested statistical methods for the conversion of, e.g., interquartile range (IQR) or median values. The PRISMA flow diagram illustrating the literature search, identification of studies, and inclusion and exclusion criteria is shown in Figure 2. For the meta-analysis of each element, we examined full-length research articles and covered the following studies and time periods: 23 studies (1982–2024) for Zn (µg/L), 16 studies (2005–2023) for Fe (µg/L), 6 studies (2005–2024) for Co (µg/L), 22 studies (1982–2024) for Cu (µg/L), 7 studies (2005–2024) for Mn (µg/L), and 10 studies (1976–2024) for Se (µg/L). In total, we included 84 studies (1976–2024) in our meta-analysis.

### 2.3. Study Selection and Data Extraction

Two trained researchers (A.S. and S.P.) independently participated in the data extraction. The following data were extracted from each study: author(s) and year of publication, element studied, type of study, country of origin, sample size (cases/controls), mean age (cases/controls), sex (females/males in cases/controls), type of clinical material (serum or plasma), and element level (mean (µg/L) ± SD, reported for cases and controls) (Table 1). In cases where results were reported as mean ± SEM, the SEM (standard error of the mean) was converted to SD using the appropriate formula. When the authors reported the results as IQR, we converted the IQR to SD [35,36]. After the selection and data extraction processes, the final list of studies was compiled by consensus of the two researchers (A.S. and S.P).

### 2.4. Quality Assessment

The quality assessment of the studies was carried out using modified criteria [80] of the Newcastle–Ottawa Scale (NOS) [81]. The possible scores ranged from 1 to 6, and studies that scored 6 were considered to be of the highest quality and to have the lowest risk of bias. Studies scoring less than 6 were considered to be of lower quality and to have a higher risk of bias.

Some of the deficiencies identified in some studies were as follows: missing information on participants’ country of residence, small sample size, missing information on sex, missing information on age. Each of the relevant factors (type of study, country, sample size, mean age, biological matrix, and sex) carried 1 point, so the absence of any of these data led to a lower number of points. Furthermore, the number of points assigned to each study for each clinical matrix was determined. Each attribute (selection, comparability, outcome, and score) was assigned a letter (a, b, c, etc.), depending on how many criteria it fulfilled. Finally, the mean quality score was calculated for each entire meta-analysis. A detailed explanation can be found below, in Table 2.

### 2.5. Statistical Analysis

Heterogeneity of the selected studies was assessed using I-squared (I2) and the associated Cochran’s Q test [82] and τ-squared (τ2) [83,84], as we previously described in Stojsavljević et al. (2023) [85].

Effect sizes were calculated as mean differences in element levels and then converted to Hedges’s g, with adjustments to account for the influence of small sample sizes [86]. We also calculated 95% confidence intervals (CIs), the relative weight of each study, and the standard residual, as described elsewhere [85]. Significance was set at a two-sided *p* < 0.05.

Statistical analysis was performed using Comprehensive Meta-Analysis software (v. 3.0, Biostat Inc., Frederick, MD, USA). We used VOSviewer 1.6.19 software to generate network data maps for visualization and exploration.

### 2.6. Publication Bias

To assess publication bias and the bias of each element visually represented using funnel plots, we conducted Egger’s regression test [87] and Begg and Mazumdar’s rank correlation test [88].

### 2.7. Registration of Meta-Analysis

The current meta-analysis was registered in the International Prospective Register of Systematic Reviews (PROSPERO), powered by the National Institute for Health and Care Research (NIHR), Centre for Reviews and Dissemination, University of York, UK. The ID of this meta-analysis is as follows: CRD42024524428 (https://www.crd.york.ac.uk/PROSPERO/#myprospero, accessed on 25 March 2024).

## 3. Results

### 3.1. Selection and Identification of Studies

The process for the study selection and identification is illustrated in Figure 2. In our initial literature search, we found a total of 57,292 records consisting only of the keyword “multiple sclerosis and trace elements.” After removing 1429 duplicate records, we obtained records based on title and abstract (57,292). We then excluded another 35,017 records with irrelevant topics. We next accessed the search for the remaining 22,275 reports. Of these, 22,130 reports could not be found, and we checked the remaining 145 reports for eligibility. Of these 145 reports, 58 were missing some of the required data: in total, 12 reports were missing control data, 23 reports were review articles, and 8 reports contained extremely abnormal data. This process led to a selection of 87 studies that were included for in-depth analysis. After a final review, three of these studies were found to contain abnormal data and were therefore also excluded. Thus, the meta-analysis included a total of 84 studies. The total number of cases in all the studies was 4461, while the total number of controls participating in the studies was 3888. This resulted in a cumulative total of 8349 participants included in the meta-analysis. For each trace element tested, the cases/control ratios were as follows: Zn 1308/1096, Fe 609/595, Co 1143/1031, Cu 1143/1031, Mn 460/453, and Se 355. The total numbers of participants for each element (cases + controls) were as follows: Zn 2404, Fe 1204, Co 763, Cu 2147, Mn 913, and Se 891.

### 3.2. Characteristics of the Studies

An overview of the characteristics of the studies included in the meta-analysis is shown in Table 1. A total of 23 studies were included for the meta-analysis of Zn [37,38,39,40,41,42,43,44,45,46,47,48,49,50,51,52,53,54,55,56,57,58,59].

Twelve studies were conducted in Europe [38,41,44,45,48,50,54,55,56,57,58,87], with a further three in the USA [39,40,42], one in Africa [53], one in South America [52], and six in Asia [43,46,47,49,51,59]. A total of 16 studies were included for the meta-analysis of Fe [44,50,51,56,57,58,60,61,62,63,64,65,66,67,68]. From Europe, 12 studies were included [44,50,56,57,58,61,62,63,64,65,67,68], 2 studies were from Africa [60], and 2 studies from Asia [51,66]. The meta-analysis of Co was based on 6 studies [37,44,50,56,57,69], all from Europe. The meta-analysis of Cu was conducted based on 22 studies [37,38,41,42,43,44,45,46,48,49,50,55,56,57,58,59,65,69,70,71,72,73]. There were 15 studies from Europe [37,38,41,44,45,48,50,55,56,57,58,65,69,70,73], 1 from the USA [42], and 6 studies from Asia [43,45,49,59,71,72]. The meta-analysis of Mn was conducted on the basis of 7 studies from Europe [37,44,50,56,57,69,72]. The data from 10 studies were collected for the meta-analysis of Se [37,42,44,48,74,75,76,77,78,79]. Eight studies were from Europe [37,44,48,74,75,76,77,79], one study was from the USA [42], and one study was from Asia [78]. If we consider all 84 studies used in this meta-analysis, it is interesting to note that 71.43% of the studies were from Europe, 17.86% from Asia, 5.95% from the USA, 3.57% from Africa, and 0.012% from South America.

### 3.3. Evaluation of the Quality Assessment

The quality ratings of the studies included in the meta-analysis ranged from 1 to 6, with an average rating of 5.48. The average quality rating for the Zn studies was 5.39, for the Fe studies it was 5.62, for the Co studies it was 5.83, for the Cu studies it was 5.41, for the Mn studies it was 5.86, and for the Se it was studies 4.80 (Table 2). A score of 4 was generally assigned if the sex and age of the participants were not provided. A score of 5 was given to studies that were not case–control studies (and were therefore cohort or cross-sectional studies) or that did not report the gender or age of participants. These scores did not mean the selected studies were of lower quality, but rather indicated that they did not fully meet the criteria set for this meta-analysis.

### 3.4. Meta-Analysis of Serum/Plasma Zn Levels

The meta-analysis of the serum/plasma Zn levels included 23 studies, with a combined sample size of 1308 cases and 1096 controls (Table 1). Seven studies did not report complete data on the mean ages of the cases/controls, and four studies did not report data on the sex (female/male) of the cases/controls. In the remaining studies, the average age of the cases was 36.23 years and that of the controls 38.75 years. Sex was not reported in three studies, while in one study, sex was only reported for the cases. In the remaining studies, there were 786 females and 406 males in the case group and 584 females and 404 males in the control group.

The mean serum/plasma Zn levels ranged from 333 ± 46 µg/L [44] to 2615 ± 628 µg/L [48] for the case groups and from 340 ± 59 µg/L [44] to 2811 ± 569 µg/L [48] for the control groups. Of the 23 studies, three reported significantly higher Zn levels in the case groups than in the control groups, ten reported significantly lower Zn levels in the case groups than in the control groups, and ten reported no significant differences in Zn levels between the two groups.

Pooling the data using the random-effects model revealed cases had significantly lower Zn levels than controls, with Hedges’s g = 0.694 (95% CI: 155, 1233) and *p* = 0.012 (Figure 3). The effect sizes of the individual studies ranged from −6.050 (95% CI: −6.975, −5.126, *p* = 0.000) [49] to 4.206 (95% CI: 3.678, 4.734, *p* = 0.000) in the study by Alimonti et al. (2007) [50]. The relative weights and standard residuals for each study are shown in Figure 3. The relative weights ranged from 4.04% [49] to 4.54% [52]. The standard residuals ranged from −5.03 [49] to 2.74 [50].

High heterogeneity was observed with I^2^ = 96.843%, Q(23) = 694.857 and τ^2^ = 1.650, *p* = 0.012, indicating substantial variation in true mean effects between the studies. Publication bias was assessed using funnel plots, which indicated no significant publication bias. Egger’s regression test yielded t_22_ = −3.231, *p* = 0.313, and the Begg and Mazumdar rank correlation yielded Kendall’s τ = −0.095, *p* = 0.526 (Figure 4).

Considering all the data, the pooled effect size indicated significantly lower serum/plasma Zn levels in the MS cases than in the controls (*p* = 0.012).

### 3.5. Meta-Analysis of Serum/Plasma Fe Levels

The meta-analysis of serum/plasma Fe levels included 16 studies, with a total sample size of 609 MS patients and 595 controls (Table 1). In two studies, the mean age of the controls was not reported, and in one study, the mean ages were between 18 and 58 and 20 and 60 for the cases and controls, respectively. In the remaining studies, the mean ages were 38.82 years for the cases and 37.07 years for the controls.

One study referred only to females in both case and control groups, another study did not specify the sex of either group, and one study did not specify the sex of the control group. Two other studies included only females in the case group, as did one study in the control group. In the other studies combined, the case group consisted of 311 females and 207 males, while the control group included 272 females and 257 males (Table 1).

The mean Fe levels in the serum/plasma varied widely. In the MS case groups, they ranged from 538 ± 110 µg/L [67] to 1762 ± 747 µg/L [58]. In the control groups, the serum Fe levels ranged from 595 ± 47 µg/L [60] to 1707 ± 548 µg/L [56]. Two studies reported significantly higher Fe levels in the case groups than in the control groups, seven studies reported significantly lower Fe levels in the case groups than in the control groups, while seven studies found no significant differences between the two groups.

The forest plot of the pooled data under the random-effects model (Figure 5) showed significant differences between the cases and the controls, with Hedges’s g = 1.080 (95% CI: 0.294, 1.866) and *p* = 0.007. The effect sizes in the individual studies ranged from −0.897 (95% CI: −1.607, −1.187, *p* = 0.013) in the study by Ay et al. (2023) [58] to 4.814 (95% CI: 4.234, 5.394, *p* = 0.000) in the study by Alimonti et al. (2007) [50]. The relative weights and standard residuals for each study are shown in Figure 5. The relative weights ranged from 5.89% [66] to 6.41% [65], and the standard residuals ranged from −1.27 [58] to 3.41 [66]. High heterogeneity was observed, with I^2^ = 96.947%, Q(16) = 491.351, and τ^2^ = 2.468 with *p* = 0.007, indicating substantial variation in true mean effects between the studies.

The funnel plots indicated that no significant publication bias was present. Egger’s regression test yielded t_15_ = 5.273, *p* = 0.323, and the Begg and Mazumdar rank correlation yielded Kendall’s τ = −0.235, *p* = 0.207 (Figure 6).

In summary, the pooled effect size indicated significantly lower serum/plasma Fe levels in the MS cases than in the controls (*p* = 0.007).

### 3.6. Meta-Analysis of the Serum/Plasma Co Levels

The meta-analysis of serum/plasma Co levels in cases and controls included six studies, with a combined sample size of 405 cases and 358 controls (Table 1). The mean age was 31.83 years for the cases and 33.17 years for the controls. In one study, the number of females and males in the control group was not reported, and in another study, both the cases and the controls were exclusively female. The MS case group consisted of 136 females and 161 males, while the control group consisted of 150 females and 148 males.

The mean serum/plasma Co levels ranged from 0.07 ± 0.014 µg/L [44] to 0.574 ± 0.238 µg/L [37] in the case groups and from 0.066 ± 0.007 µg/L [44] to 1.316 ± 0.630 µg/L [37] in the control groups. One study reported a significantly higher serum Co level in the case group than in the control group, four studies found significantly lower levels in the case groups than in the control groups, and one study reported no significant differences in Co levels between the case group and the control group (Table 1).

The forest plot of the pooled data under the random-effects model is shown in Figure 7. The results show no significant differences between the two groups, with Hedges’s g = 0.666 (95% CI: −0.662, 1.994) and *p* = 0.326. The effect sizes in the individual studies ranged from −1.325 (95% CI: −1.719, −0.932, *p* = 0.000) in the study by Forte et al. (2005) [69] to 7.876 (95% CI: 5.409, 10.342, *p* = 0.000) in the study by Gellein et al. (2008) [44], indicating large heterogeneity. The relative weights and standard residuals for each study are shown in Figure 7. The relative weights ranged from 11.17% [44] to 18.02% [37], and the standard residuals ranged from −1.37 [69] to 3.77 [44]. The heterogeneity was I^2^ = 98.205%, Q(6) = 278.494 and τ^2^ = 2.528, *p* = 0.326, indicating no large heterogeneity (Figure 7).

The funnel plots (Figure 8) revealed no significant publication bias. Egger’s regression test yielded t_5_ = 0.583, *p* = 0.944, and the Begg and Mazumdar rank correlation yielded Kendall’s τ = 0.200, *p* = 0.707.

In summary, the pooled effect size showed no significant differences in Co serum/plasma levels between the MS cases and the controls (*p* = 0.326).

### 3.7. Meta-Analysis of Serum/Plasma Cu Levels

The meta-analysis of serum/plasma Cu levels included 22 studies, with a total sample size of 1143 cases and 1031 controls (Table 1). Six studies did not include age data, and one study only included age data for the cases. The mean ages in the remaining studies were 37.06 years for the cases and 33.27 years for the controls. Two studies did not report gender, two studies reported gender only for the cases, and one study included only females in the case group. The remaining case group consisted of 683 females and 377 males, while the control group consisted of 480 females and 367 males.

The mean serum/plasma Cu levels ranged from 487 ± 117 µg/L [44] to 1882 ± 82 µg/L [49] in the case groups and from 478 ± 133 µg/L [44] to 1735 ± 46 µg/L [42] in the control groups. Eight studies reported significantly higher Cu levels in the case groups than in the control groups, four studies reported lower Cu levels in the case groups, while no significant differences between the two groups were observed in ten studies (Table 3).

The forest plot of the pooled data under the random-effects model (Figure 9) showed significant differences between the cases and the controls, with Hedges’s g = −0.833 (95% CI: −1.441, −0.325) and *p* = 0.002. The effect sizes in the individual studies ranged from −12.768 (95% CI: −14.580, −10.956, *p* = 0.000) in the study by Ghoreishi et al. (2015) [49] to 1.095 (95% CI: 0.437, 1.752, *p* = 0.001) in the study by Kapaki et al. (1989) [41]. The relative weights and standard residuals for each study are shown in Figure 9. The relative weights ranged from 3.20% [49] to 4.78% [37] and the standard residuals ranged from −7.59 [49] to 1.51 [41]. The heterogeneity was high, with I^2^ = 96.895%, Q(22) = 676.382 and τ^2^ = 1.681, *p* = 0.002.

The funnel plots revealed significant publication bias. Egger’s regression test yielded t_21_ = −7.244, *p* = 0.021, and the Begg and Mazumdar rank correlation yielded Kendall’s τ = −0.363, *p* = 0.018 (Figure 10).

In summary, the pooled effect size indicated significantly higher serum/plasma Cu levels in the MS cases than in the controls (*p* = 0.002).

### 3.8. Meta-Analysis of Mn Serum/Plasma Levels

The meta-analysis of serum/plasma Mn levels included seven studies, with a total sample size of 460 cases and 453 controls (Table 1). The mean ages were 37.29 years for the cases and 33.57 years for the controls. In one study, the MS group consisted only of females, and one study provided no data on the sex of the control group. The MS group consisted of 291 females and 169 males, and the control group consisted of 217 females and 176 males.

The mean serum/plasma Mn levels ranged from 0.299 ± 0.284 µg/L [37] to 8.39 ± 3.93 µg/L [69] in the case groups and from 0.57 ± 0.17 µg/L [56] to 7.80 ± 3 µg/L [69] in the control groups. Three studies reported significantly higher Mn levels in the case groups than in the control groups, three studies reported significantly lower Mn levels in the case groups than in the control groups, and one study found no significant difference between the two groups.

The forest plot of the pooled data under the random-effects model is shown in Figure 11. The results showed no significant differences between the two groups, with Hedges’s g = 0.369 (95% CI: −0.975, 1.714) and *p* = 0.590. The effect sizes in the individual studies ranged from −1.862 (95% CI: −2.797, −0.927, *p* = 0.000) in the study by Visconti et al. (2005) [57] to 2.683 (95% CI: 2.367, 2.999, *p* = 0.000) in the study by Stojsavljević et al. (2024) [37]. The relative weights and standard residuals for each study are shown in Figure 11. The relative weights ranged from 13.69% [57] to 14.54% [37,50]. The standard residuals ranged from −1.30 [57] to 1.39 [37]. Heterogeneity was present, with I^2^ = 98.567%, Q(7) = 418.634 and τ^2^ = 3.209 (*p* = 0.590), indicating a high degree of variation between the studies.

The funnel plots (Figure 12) showed no significant publication bias. Egger’s regression test yielded t_6_ = −3.218, *p* = 0.769, and the Begg and Mazumdar rank correlation yielded Kendall’s τ = 0.000, *p* = 1.000.

In summary, the pooled effect size did not reveal significantly different serum/plasma Mn levels in the MS cases or in the controls (*p* = 0.590).

### 3.9. Meta-Analysis of Serum/Plasma Se Levels

The meta-analysis of serum/plasma Se levels included ten studies, with a total sample size of 536 cases and 355 controls (Table 1). Four studies did not include age data, and one study did not include age data for the controls. The mean age was 42 years for the cases and 41.60 years for the controls. No data were included on sex in four studies for either cases or controls and in one study, no data were included for controls. The case group consisted of 135 females and 63 males, and the control group consisted of 76 females and 56 males.

The mean serum/plasma Se levels ranged from 45.40 ± 2.10 µg/L [44] to 123 ± 17 µg/L [75] in the case groups and from 41.1 ± 2.2 µg/L [44] to 123 ± 60 µg/L [78] in the control groups. Two studies reported significantly higher Se levels in the case than in the control groups, four studies reported significantly lower levels in the case groups than in the control groups, and no significant differences between the two groups were found in two studies.

The forest plot of the pooled data under the random-effects model is shown in Figure 13. The results showed no significant differences between the two groups, with Hedges’s g = 0.253 (95% CI: −0.278, 0.784) and *p* = 0.351. The effect sizes in the individual studies ranged from −1.915 (95% CI: −0.920, *p* = 0.000) in the study by Gellein et al. (2008) [44] to 1.822 (95% CI: 1.223, 2.420, *p* = 0.000) in the study by Smith et al. (1989) [42]. The relative weights and standard residuals for each study are shown in Figure 13. The relative weights ranged from 8.25% [44] to 11.32% [37]. The standard residuals ranged from −2.40 [44] to 1.40 [77]. Heterogeneity was present, with I^2^ = 91.474%, Q(10) = 105.561 and τ^2^ = 0.633 (*p* = 0.351), indicating a high degree of variation between the studies.

The funnel plots (Figure 12) showed no significant publication bias. Egger’s regression test yielded t_9_ = −4.292, *p* = 0.089, and Begg and Mazumdar’s rank correlation yielded Kendall’s τ = −0.400, *p* = 0.107 (Figure 14).

In summary, the pooled effect size did not reveal significantly different serum/plasma Se levels in the MS cases or in the controls (*p* = 0.351).

Overall, this meta-analysis revealed significantly lower serum/plasma Zn and Fe levels and higher serum/plasma Cu levels in the MS-affected individuals compared to the controls. At the same time, no significant differences were found between the MS cases and the controls with regards to serum/plasma levels of Co, Mn, or Se. A statistically significant publication bias was only found in the meta-analysis of Cu in the serum/plasma of the MS cases and controls.

## 4. Discussion

Zn is a redox-inactive trace element that plays catalytic, structural, and regulatory roles. It is the second most abundant transition metal in the brain, mainly located in presynaptic vesicles in the neurons of the cerebral cortex, amygdala, and hippocampus [89,90]. The brain of an adult human contains about 40 mg of Zn [26]. Due to its antioxidant, anti-inflammatory, and neuroprotective properties, Zn is crucial for cell growth, cell proliferation, synaptic transmission, and other biochemical pathways in the brain [24,91]. Approximately 10% of the human proteome binds Zn [92]. Thus, homeostasis of Zn in the brain must be maintained, otherwise a deficiency can occur [93]. According to the World Health Organization (WHO), Zn deficiency is widespread worldwide, and it is ranked as the 5th as leading health-risk factor in affluent countries [94,95]. Elitt et al. (2019) [96] reported a Zn decline in white matter lesions and myelin loss in MS, whereas Bredholt and Frederiksen’s [97] meta-analysis (2016) found significantly higher Zn levels in cerebrospinal fluid (CSF) from MS patients than in that from controls. In addition, most of the studies in our meta-analysis showed lower serum/plasma Zn levels in MS patients than in controls (Table 1). Bredholt and Frederiksen (2016) [97] included 13 studies in their meta-analysis (705 cases and 739 controls) and found significantly lower serum/plasma Zn levels in the MS patients than in the controls. Nirooei et al. (2022) [98] reviewed 17 studies in their meta-analysis and reported significantly lower serum/plasma Zn levels in MS cases than in controls. The current meta-analysis, in which serum/plasma Zn levels were compared in a considerably greater number of participants (1308 cases and 1096 controls), is in good agreement with these previous meta-analyses on circulating body fluids. Overall, these findings highlight the need to monitor Zn levels in MS patients’ serum/plasma and give further insight into modifying the diet or implementing appropriate supplemental therapy. Furthermore, studies conducted about five decades ago indicated the importance of monitoring serum Zn levels in MS patients, particularly depending on age, sex, and other demographic subgroups [38]. However, these trends have not been supported over time. The latest study, conducted by Stojsavljević et al. (2024) [37], showed that MS patients had significantly lower serum Zn levels than matched controls based on sex, age, and smoking habits. That study also reported a positive correlation between Zn levels and the Expanded Disability Status Scale (EDSS), indicating the need to monitor the status of Zn and other essential trace elements in serum/plasma, depending on disease course, progression, and other clinical variables.

Fe is an essential metal for brain homeostasis, participating in mitochondrial respiration, neurotransmitter synthesis/metabolism, and myelin synthesis, among other pathways [99]. Compared to other nerve cells, oligodendrocytes contain the most Fe [100]. It has been suggested that with aging, the accumulation of Fe in the substantia nigra, putamen, caudate nucleus, globus pallidus, and cortices could lead to neurodegenerative processes [101]. Higher Fe levels occur in specific brain regions in MS, most prominently in the deep structures of the gray matter. Furthermore, as MS advances, Fe levels increase in many deep gray matter areas, while Fe deposits in MS patients’ white matter were frequently located in sites of inflammation [30]. On the other hand, Fe deficiency is the most frequent nutritional deficiency worldwide, even in developed countries, with nearly 30% of the population, primarily women, afflicted by this condition [102,103]. Iron deficiency was studied in depth in a recent review by Kumar et al. (2022) [104]. The current meta-analysis showed lower serum/plasma Fe levels in MS patients than in controls (Table 1 and Table 3). These findings indicate the importance of monitoring Fe in the bloodstreams of MS patients, as well as providing a rational basis for considering a change in diet and/or supplementation.

Cobalt constitutes a part of cobalamin (vitamin B12), and this role makes it an essential trace element [105]. Cobalamin deficiency leads to pernicious and megaloblastic anemia, as well as demyelination, axonal degeneration, and inflammation in MS [106,107]. A meta-analysis by Zhu et al. (2011) [108] showed significantly lower serum/plasma levels of cobalamin in MS patients than in controls, while a meta-analysis by Dardiotis et al. (2017) [109] showed no differences in serum/plasma cobalamin levels between MS cases (n = 767) and controls (n = 762). These contradictory findings necessitate additional research. In relation to cobalamin, the levels of Co in MS have not been sufficiently investigated. Most (4/6) of the studies we enrolled found higher serum/plasma Co levels in MS patients than in controls (Table 1). The most comprehensive analysis of the importance of Co in MS was provided by Stojsavljević et al. (2024) [37]. The authors showed differences in serum Co levels in MS patients and controls, but also in serum Co levels between sexes, smoking habits, and age groups of MS patients compared to their relative controls. They also reported that serum Co levels significantly distinguished RRMS from PPMS, suggesting that further investigation into the exact role of Co in MS is warranted.

Copper is a redox-active trace metal, necessary for the activation of neuropeptides, synthesis of neurotransmitters, myelination of neurons, etc. [25,110]. The brain contains about 9% of the body’s Cu, which is the third highest level of Cu in any organ [111]. Generally, higher levels of Cu and Fe are found in those parts of the brain that are depleted of Zn; for example, higher Cu levels were detected in the cortex and the white matter of the cerebellum than in cerebrum [112]. Both Cu deficiency and toxicity have serious consequences for neurological health [112]. High levels of Cu are harmful, since they promote a Fenton-like reaction, leading to the production of reactive oxygen species. The role of Cu in MS has not been fully elucidated [113], although De Riccardis et al. (2018) [70] reported significantly higher Cu levels in the CSF from patients with MS compared to that from controls. A recent meta-analysis by Nirooei et al. (2022) [98] showed that circulating Cu levels in MS patients and controls were similar. The authors included 11 studies (507 cases and 540 controls). These data contradict the meta-analysis by Sarmadi et al. (2020) [114], who reported higher serum/plasma Cu in MS cases (n = 797) than in controls (n = 875), and our current meta-analysis, which enrolled 22 studies (1143 cases and 1031 controls), found higher serum/plasma Cu levels in MS patients than in controls. In addition, a recent study by Stojsavljević et al. (2024) [37] demonstrated the ability of serum Cu to differentiate SPMS (852 ± 195 µg/L) from PPMS (1033 ± 186 µg/L). Although statistically significant, the values for SPMS and PPMS were within the widely established reference range (700–1400 µg/L); therefore, caution is advised when interpreting the data. The same authors found a negative correlation of Cu with the Multiple Sclerosis Severity Score (MSSS). These discrepancies in findings highlight the need for further research to unravel the exact roles/mechanisms of Cu in MS.

Manganese is a cofactor or activator of metalloproteins and can be found in the human body as Mn^2+^ and Mn^3+^ [115]. Due to its valence states, Mn interferes with Fe metabolism (Fe deficiency usually leads to Mn toxicity) [116]. Manganese accumulates in specific parts of the brain and, thus, shows high selectivity, particularly for the Fe-rich brain parts. Under physiological conditions, human brain Mn levels range from 5.32 to 14.0 ng Mn/mg protein; the pathophysiological threshold was estimated in the range of 16.0–42.1 ng Mn/mg protein. Natural Mn deficiency has never been reported in humans. In contrast, excessive exposure to Mn leads to neurotoxicity (manganism, a Parkinson-like syndrome) [29,117]. The role of Mn in MS has not been sufficiently studied. Melo et al. (2003) [113] and De Riccardis et al. (2018) [70] reported significantly lower Mn levels in the CSF of MS patients than the levels in healthy controls. The study conducted by Stojsavljević et al. (2024) [37] showed significantly lower serum Mn levels in MS patients than in controls, as well as significantly lower Mn levels in women, men, nonsmokers, smokers, and different age groups (20–40 and 41–60 years) with MS compared to their respective healthy controls. They also found a positive correlation of serum Mn with EDSS. Although the difference was not significant, they reported that patients with RRMS had lower serum Mn levels than those with a progressive disease course [37]. Overall, although our current meta-analysis showed no important differences between cases and controls regarding serum/plasma Mn levels, primarily due to the small number of studies conducted over decades (we were able to enroll six studies only), all the findings should be further considered, since Mn could be an important trace metal in MS due to its unique neurobiochemistry.

Selenium primarily plays antioxidant and anti-inflammatory roles in the brain [118]. Ramos et al. (2014) [119] found the highest concentrations of Se in the putamen, parietal inferior lobule, and occipital cortex and the lowest Se levels in the medulla and cerebellum. Selenium has a narrow range between essentiality and toxicity [120]. In addition to Fe and Zn, Se deficiency is more common than deficiencies in other essential trace elements [121,122], and new and valuable details can be found in the review by Shreenath et al. (2024) [123]. On the other hand, the role of Se in MS has not been sufficiently studied. Of the eight studies included in this meta-analysis, four reported lower Se values in cases than in controls, two found higher Se levels in cases than in controls, and two studies reported no statistically significant differences in Se levels in cases and controls. Our results, showing that serum Se levels in MS patients were similar to the Se levels in controls, were consistent with recent meta-analyses by Nirooei et al. (2022) [98] and Zhou et al. (2023) [124]. However, both these meta-analyses included only a relatively small number of participants, which is why the further evaluation of the role of Se in MS is necessary. Interesting findings regarding Se were presented by Stojsavljević et al. (2024) [37]. They found the ability of Se to stratify mild RRMS (71 ± 18 µg/L) from very severe SPMS (60 ± 15 µg/L); the patients with very severe SPMS had the lowest serum Se levels (87 ± 12 µg/L) compared to the controls. The authors also found differences in Se levels between the MS patients and the controls that depended on demographic subgroups, which should motivate researchers worldwide to look more clearly at the importance of Se in this complex neurological disease.

## 5. Conclusions

The data in this meta-analysis indicated changes in the levels of some essential trace elements in the serum/plasma of patients with MS. This meta-analysis revealed significantly lower serum/plasma Zn and Fe levels and higher Cu levels in MS cases than in controls. At the same time, no significant differences were found for serum/plasma levels of Co, Mn, or Se in these two groups. Therefore, the loss of circulatory Fe and Zn should be considered in supplementation/nutrition strategies for MS patients. On the other hand, since high Cu levels indicate a burden on the bloodstreams of MS patients, this trace metal should be excluded from these patients’ supplement strategies. Furthermore, all three trace elements (Fe, Zn, and Cu) should be considered from an etiological point of view and, most importantly, their levels in the bloodstreams of MS patients should be monitored. Overall, this study highlights the necessity of further research, paving the way for personalized and targeted strategies in the management of MS.

## Figures and Tables

**Figure 1 biomedicines-12-01589-f001:**
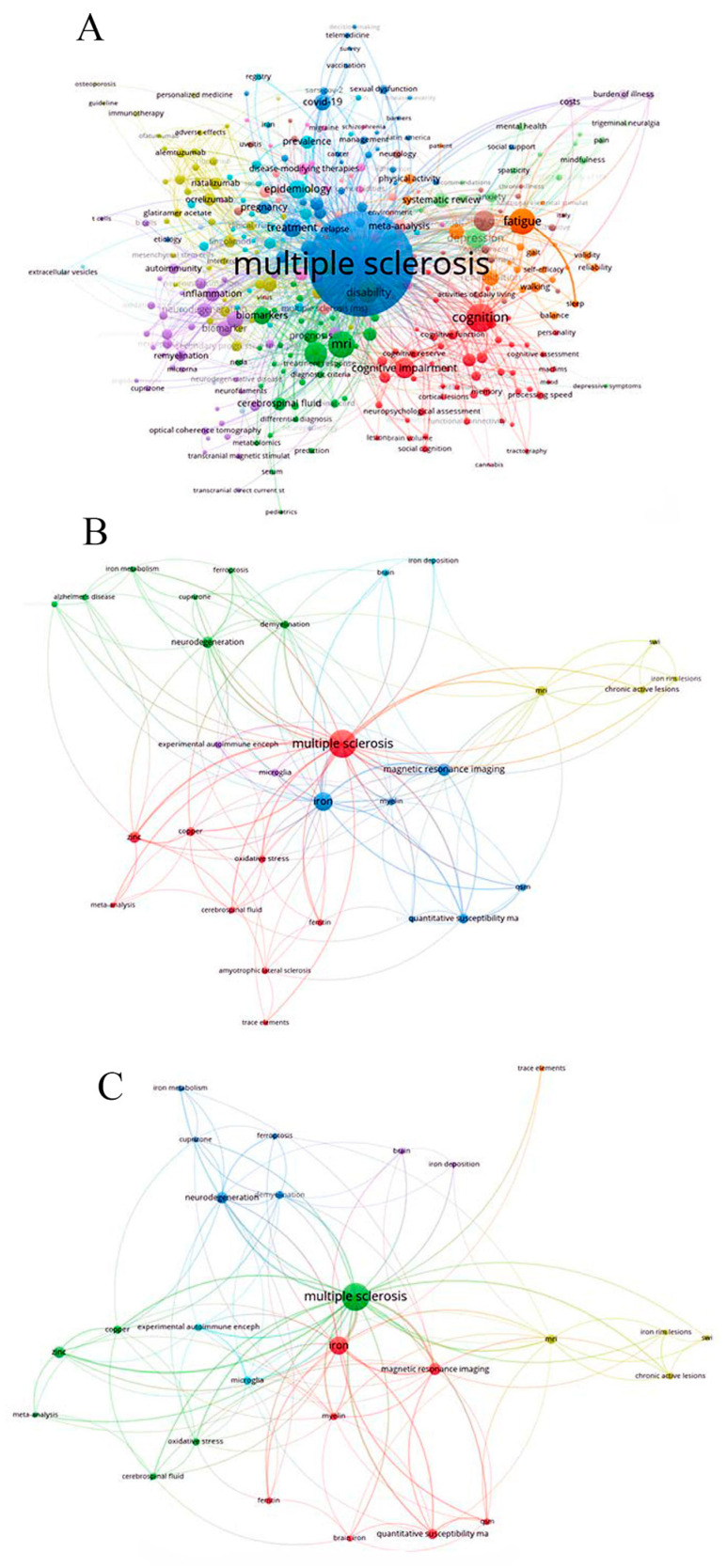
Focused Comprehensive Network Visualization of: (**A**). a total of 5051 registered multiple-sclerosis-related author keywords in scholarly literature; (**B**). 672 detected multiple sclerosis and trace-element-related author keywords in scholarly literature; (**C**). 27 most commonly used multiple-sclerosis- and trace-element-related author keywords in scholarly literature.

**Figure 2 biomedicines-12-01589-f002:**
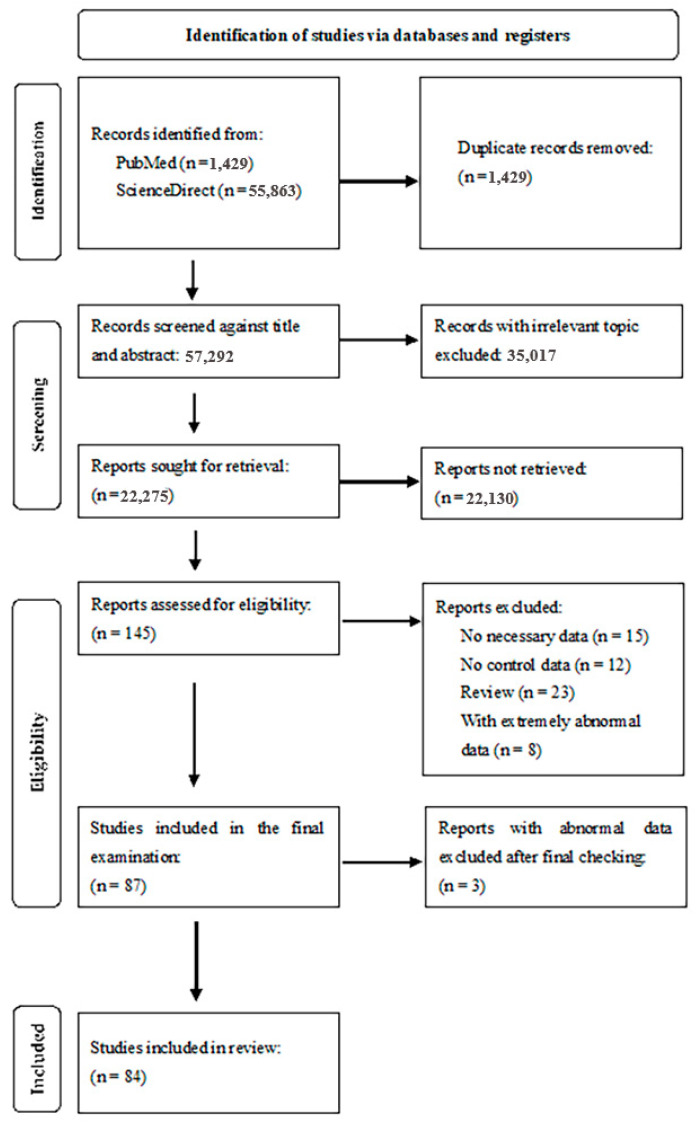
PRISMA flow diagram illustrating literature search, study identification, inclusion, and exclusion process. Abbreviation: n (number of studies).

**Figure 3 biomedicines-12-01589-f003:**
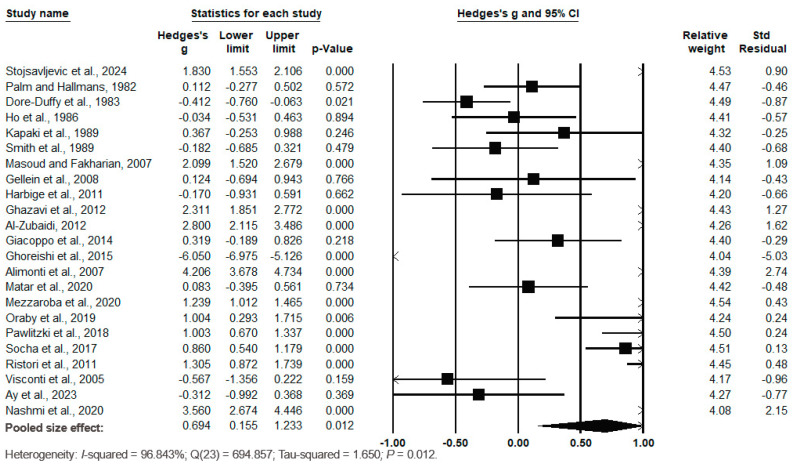
Forest plot for random-effects meta-analysis. Differences in serum/plasma Zn levels between controls and cases are shown. The size of each square is proportional to the weight of the study. The diamond symbol indicates the pooled total effect size for the set of studies included in the meta-analysis. Abbreviation: CI = confidence interval [37,38,39,40,41,42,43,44,45,46,47,48,49,50,51,52,53,54,55,56,57,58,59].

**Figure 4 biomedicines-12-01589-f004:**
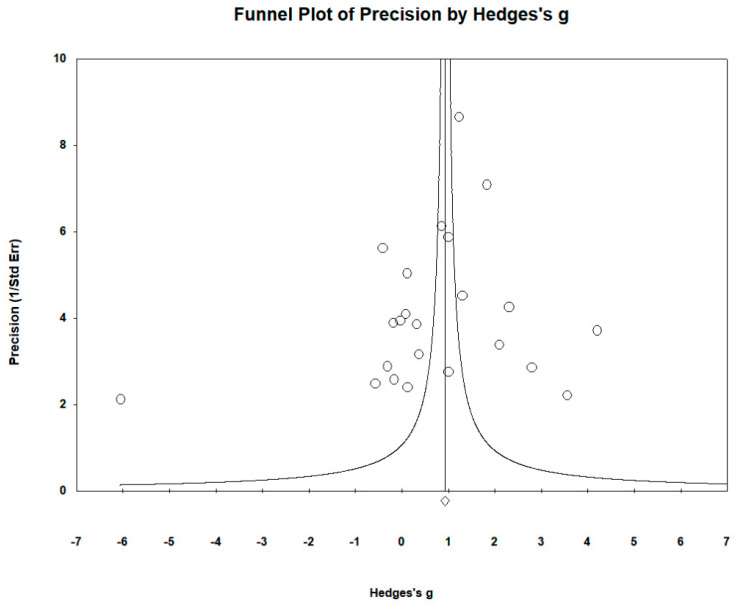
Funnel plots to assess publication bias in observed studies comparing Zn levels in the serum/plasma of controls and cases. The figure shows the effect size (Hedges’s g) of the studies against their precision (inverse of SE). The circles represent observed studies. The diamond symbol indicates the pooled overall effect size based on the observed studies. Egger’s regression test: t_22_ = −3.231, *p* = 0.313; Begg and Mazumdar rank correlation: Kendall’s τ = −0.095, *p* = 0.526.

**Figure 5 biomedicines-12-01589-f005:**
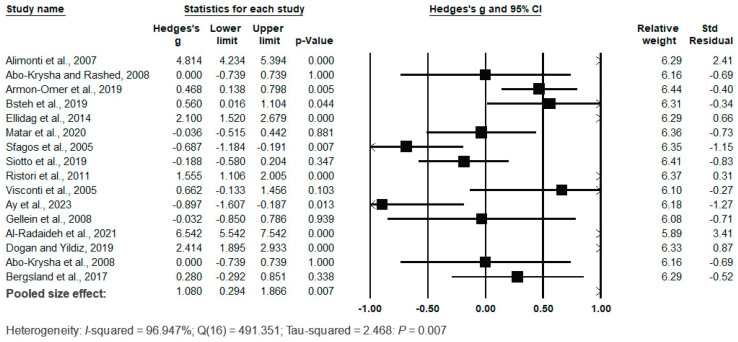
Forest plot for random-effects meta-analysis. Differences in serum/plasma Fe levels between controls and cases are shown. The size of each square is proportional to the weight of the study. The diamond symbol indicates the pooled total effect size for the set of studies included in the meta-analysis. Abbreviation: CI = confidence interval [44,50,51,56,57,58,60,61,62,63,64,65,66,67,68].

**Figure 6 biomedicines-12-01589-f006:**
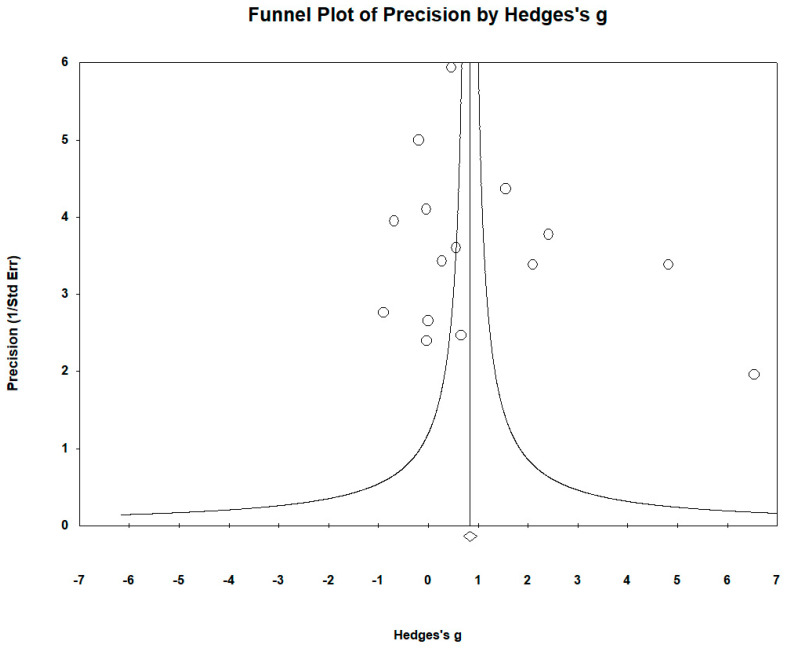
Funnel plots to assess publication bias in observed studies comparing Fe levels in the serum/plasma of controls and cases. The figure shows the effect size (Hedges’s g) of the studies against their precision (inverse of SE). The circles represent observed studies. The diamond symbol indicates the pooled overall effect size based on the observed studies. Egger’s regression test: t_15_ = 5.273, *p* = 0.323; Begg and Mazumdar rank correlation: Kendall’s τ = 0.235, *p* = 0.207.

**Figure 7 biomedicines-12-01589-f007:**
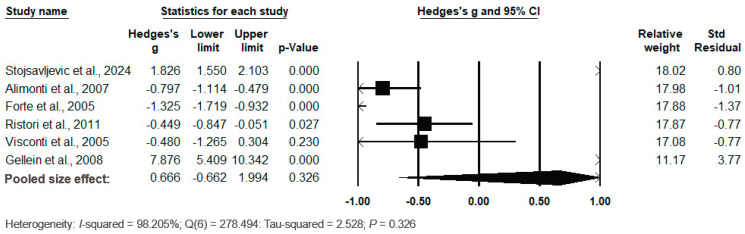
Forest plot for random-effects meta-analysis. Differences in serum/plasma Co levels between controls and cases are shown. The size of each square is proportional to the weight of the study. The diamond symbol indicates the pooled total effect size for the set of studies included in the meta-analysis. Abbreviation: CI = confidence interval [37,44,45,46,47,48,49,50,56,57,69].

**Figure 8 biomedicines-12-01589-f008:**
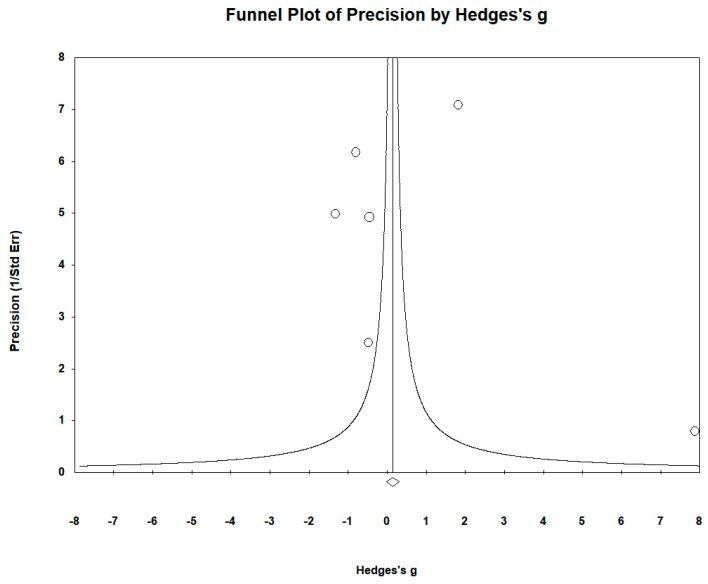
Funnel plots to assess publication bias in observed studies comparing Co levels in the serum/plasma of controls and cases. The figure shows the effect size (Hedges’s g) of the studies against their precision (inverse of SE). The circles represent observed studies. The diamond symbol indicates the pooled overall effect size based on the observed studies. Egger’s regression test: t_5_ = 0.583, *p* = 0.944; Begg and Mazumdar rank correlation: Kendall’s τ = 0.200, *p* = 0.707.

**Figure 9 biomedicines-12-01589-f009:**
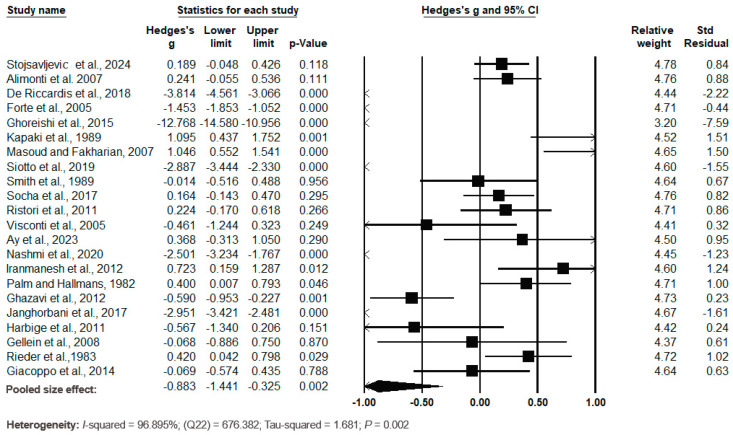
Forest plot for random-effects meta-analysis. Differences in serum/plasma Cu levels between controls and cases are shown. The size of each square is proportional to the weight of the study. The diamond symbol indicates the pooled total effect size for the set of studies included in the meta-analysis. Abbreviation: CI = confidence interval [37,38,41,46,48,49,50,55,56,57,58,59,60,65,66,67,68,69,70,71,72,73].

**Figure 10 biomedicines-12-01589-f010:**
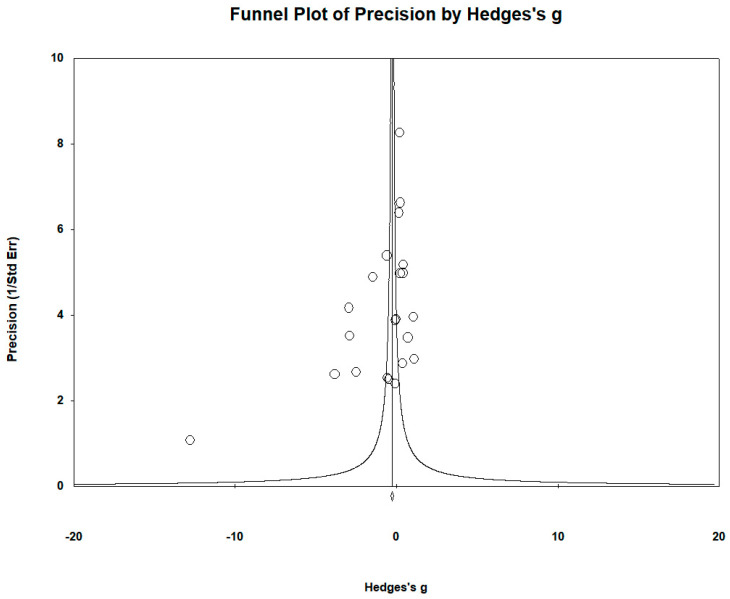
Funnel plots to assess publication bias in observed studies comparing Cu levels in the serum/plasma of controls and cases. The figure shows the effect size (Hedges’s g) of the studies against their precision (inverse of SE). The circles represent observed studies. The diamond symbol indicates the pooled overall effect size based on the observed studies. Egger’s regression test: t_21_ = −7.244, *p* = 0.021; Begg and Mazumdar rank correlation: Kendall’s τ = −0.363, *p* = 0.018.

**Figure 11 biomedicines-12-01589-f011:**
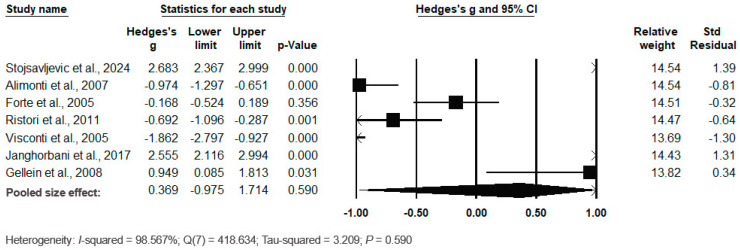
Forest plot for random-effects meta-analysis. Differences in serum/plasma Mn levels between controls and cases are shown. The size of each square is proportional to the weight of the study. The diamond symbol indicates the pooled total effect size for the set of studies included in the meta-analysis. Abbreviation: CI = confidence interval [37,44,50,56,57,69,72].

**Figure 12 biomedicines-12-01589-f012:**
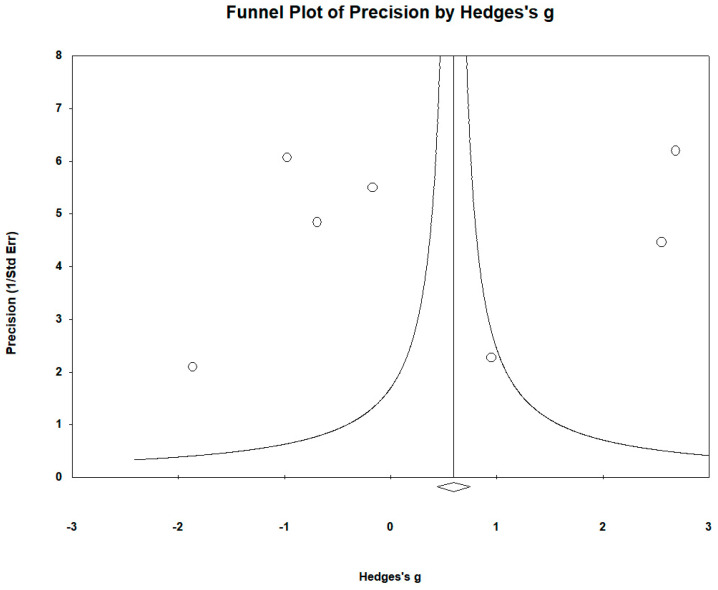
Funnel plots to assess publication bias in observed studies comparing Mn levels in the serum/plasma of controls and cases. The figure shows the effect size (Hedges’s g) of the studies against their precision (inverse of SE). The circles represent observed studies. The diamond symbol indicates the pooled overall effect size based on the observed studies. Egger’s regression test: t_6_ = −3.218, *p* = 0.769; Begg and Mazumdar rank correlation: Kendall’s τ = −0.000, *p* = 1.000.

**Figure 13 biomedicines-12-01589-f013:**
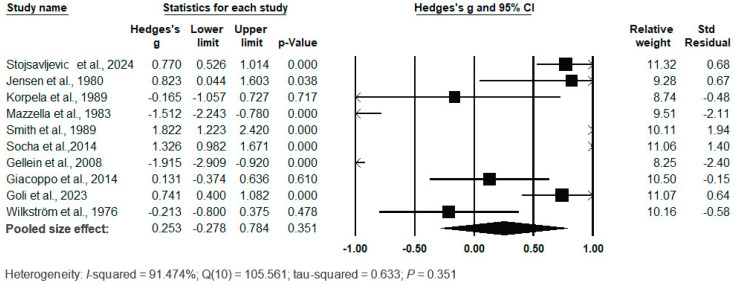
Forest plot for random-effects meta-analysis. Differences in serum/plasma Se levels between controls and cases are shown. The size of each square is proportional to the weight of the study. The diamond symbol indicates the pooled total effect size for the set of studies included in the meta-analysis. Abbreviation: CI = confidence interval [37,42,44,48,74,75,76,77,78,79].

**Figure 14 biomedicines-12-01589-f014:**
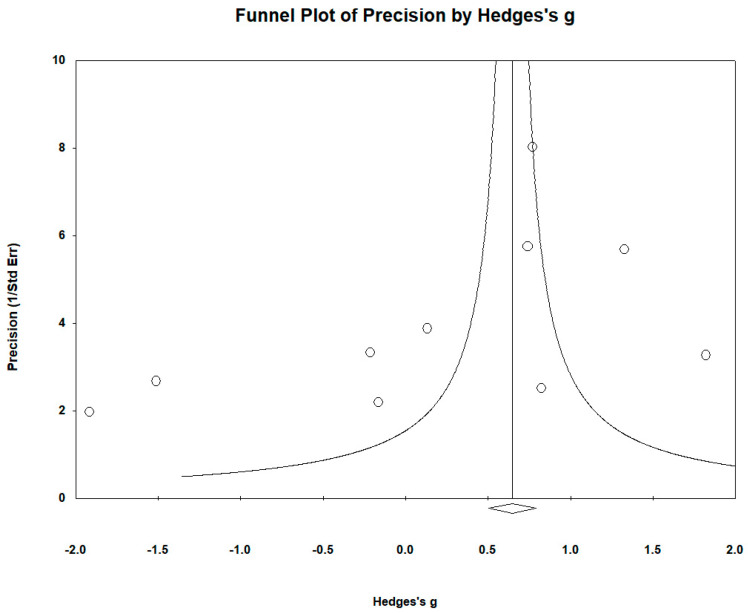
Funnel plots to assess publication bias in observed studies comparing Se levels in the serum/plasma of controls and cases. The figure shows the effect size (Hedges’s g) of the studies against their precision (inverse of SE). The circles represent observed studies. The diamond symbol indicates the pooled overall effect size based on the observed studies. Egger’s regression test: t_9_ = −4.292, *p* = 0.089; Begg and Mazumdar rank correlation: Kendall’s τ = −0.400, *p* = 0.107.

**Table 1 biomedicines-12-01589-t001:** Characteristics of 84 studies (1976–2024) included in the systematic review and meta-analysis of Zn, Fe, Co, Cu, Mn, and Se levels (µg/L) in the serum or plasma of patients with MS (cases) and control volunteers (controls) were as follows: 23 studies (1982–2024) included in the systematic review and meta-analysis of Zn level, 16 studies (2005–2023) included in the systematic review and meta-analysis of Fe level, 6 studies (2005–2024) included in the systematic review and meta-analysis of Co level, 22 studies (1982–2024) included in the systematic review and meta-analysis of Cu level, 7 studies (2005–2024) included in the systematic review and meta-analysis of Mn level, and 10 studies (1976–2024) included in the systematic review and meta-analysis of Se level.

Σ		Study	Element	Type of Study	Country	Sample Size Cases/Controls	Mean Age Cases/Controls	Sex: Female/Male; Cases/Controls	Biological Matrix	Element Level µg/L (Mean ± SD): Cases/Controls
1.	1.	Stojsavljević et al., 2024 [37]	Zn	Case–control	Serbia	215/100	43 ± 10/42 ± 9	120/95; 66/34	Serum	725 ± 250/1827 ± 1003
2.	2.	Palm and Hallmans, 1982 [38]	Zn	Case–control	Sweden	50/50	N.A./N.A.	29/21; 29/21	Serum	850 ± 124/863 ± 105
3.	3.	Dore-Duffy et al., 1983 [39]	Zn	Case–control	USA	68/60	N.A./N.A.	50/18; 30/30	Plasma	843 ± 157/785 ± 118
4.	4.	Ho et al., 1986 [40]	Zn	Case–control	USA	45/23	N.A./N.A.	N.A./N.A.	Plasma	889 ± 203/883 ± 98
5.	5.	Kapaki et al., 1989 [41]	Zn	Case–control	Greece	15/28	34 ± 10/46 ± 14	3/12; 10/18	Serum	1033 ± 137/1098 ± 190
6.	6.	Smith et al., 1989 [42]	Zn	Cohort	USA	27/33	N.A./N.A.	17/10; 11/22	Plasma	1026 ± 196/1000 ± 70
7.	7.	Masoud and Fakharian, 2007 [43]	Zn	Cohort	Iran	35/35	32 ± 7/35 ± 5	28/7; 28/7	Serum	856 ± 137/1098 ± 85
8.	8.	Gellein et al., 2008 [44]	Zn	Case–control	Norway	9/13	44 ± 3/49 ± 4	9/0; 8/5	Plasma	333 ± 46/340 ± 59
9.	9.	Harbige et al., 2011 [45]	Zn	Case–control	UK	21/9	N.A./N.A.	N.A./N.A.	Plasma	883 ± 163/856 ± 131
10.	10.	Ghazavi et al., 2012 [46]	Zn	Case–control	Iran	60/60	N.A./N.A.	43/17; 38/22	Serum	399 ± 321/1275 ± 425
11.	11.	Al-Zubaidi, 2012 [47]	Zn	Case–control	Iraq	32/32	32 ± 7/N.A.	24/8; N.A.	Serum	1066 ± 216/1504 ± 33
12.	12.	Giacoppo et al., 2014 [48]	Zn	Case–control	Italy	41/23	41 ± 2/35 ± 2	31/10; 14/9	Plasma	2615 ± 628/2811 ± 569
13.	13.	Ghoreishi et al., 2015 [49]	Zn	Case–control	Iran	50/50	32 ± 3/32 ± 3	N.A./N.A.	Serum	720 ± 39/460 ± 46
14.	14.	Alimonti et al., 2007 [50]	Zn	Case–control	Italy	60/124	39 ± 10/45 ± 13	38/22; 43/81	Serum	650 ± 25/795 ± 38
15.	15.	Matar et al., 2020 [51]	Zn	Case–control	Lebanon	27/42	43 ± 13/38 ± 16	14/13; 29/13	Serum	806 ± 152/820 ± 176
16.	16.	Mezzaroba et al., 2020 [52]	Zn	Case–control	Brazil	174/182	42 ± 13/40 ± 10	121/53; 128/54	Serum	1090 ± 254/1540 ± 442
17.	17.	Oraby et al., 2019 [53]	Zn	Case–control	Egypt	25/12	31 ± 9/29 ± 7	21/4; 14/11	Serum	653 ± 169/822 ± 155
18.	18.	Pawlitzki et al., 2018 [54]	Zn	Cohort	Germany	151/50	43 ± 12/43 ± 14	113/38; 38/12	Serum	817 ± 137/955 ± 137
19.	19.	Socha et al., 2017 [55]	Zn	Case–control	Poland	101/68	41 ± 10/40 ± 13	64/37; 47/21	Serum	776 ± 195/992 ± 315
20.	20.	Ristori et al., 2011 [56]	Zn	Case–control	Italy	49/49	36 ± 7/33 ± 6	29/20; 26/23	Serum	655 ± 83/808 ± 142
21.	21.	Visconti et al., 2005 [57]	Zn	Case–control	Italy	12/12	28 ± 8/28 ± 9	8/4; 7/5	Serum	864 ± 160/781 ± 120
22.	22.	Ay et al., 2023 [58]	Zn	Case–control	Türkiye	16/16	45 ± 7/44 ± 12	9/7; 11/5	Serum	1439 ± 568/1292 ± 316
23.	23.	Nashmi et al., 2020 [59]	Zn	Case–control	Iraq	25/25	42 ± 9/41 ± 10	15/10; 14/11	Serum	667 ± 76/933 ± 71
24.	1.	Alimonti et al., 2007 [50]	Fe	Case–control	Italy	60/124	39 ± 10/45 ± 13	38/22; 43/81	Serum	936 ± 117/1610 ± 149
25.	2.	Abo-Krysha and Rashed, 2008 [60]	Fe	Case–control	Egypt	20/10	30 ± 9/N.A.	Only females	Serum	595 ± 31/595 ± 47
26.	3.	Armon-Omer et al., 2019 [61]	Fe	Cross-sectional	Israel	63/83	45 ± 14/41 ± 12	42/21; 49/34	Serum	627 ± 353/787 ± 330
27.	4.	Bsteh et al., 2019 [62]	Fe	Case–control	Austria	71/16	46 ± 8/32 ± 12	N.A./N.A.	Serum	857 ± 325/1070 ± 560
28.	5.	Ellidag et al., 2014 [63]	Fe	Case–control	Türkiye	35/35	38 ± 11/38 ± 10	15/20; 22/13	Serum	673 ± 85/816 ± 43
29.	6.	Matar et al., 2020 [51]	Fe	Case–control	Lebanon	27/42	43 ± 13/38 ± 16	14/13; 29/13	Serum	847 ± 351/833 ± 397
30.	7.	Sfagos et al., 2005 [64]	Fe	Case–control	Greece	27/40	38 ± 6/N.A.	17/10; N.A.	Serum	1669 ± 478/1417 ± 258
31.	8.	Siotto et al., 2019 [65]	Fe	Case–control	Italy	60/42	37 ± 9/40 ± 11	45/15; 22/20	Serum	857 ± 430/786 ± 276
32.	9.	Ristori et al., 2011 [56]	Fe	Case–control	Italy	49/49	36 ± 7/33 ± 6	29/20; 26/23	Serum	985 ± 352/1707 ± 548
33.	10.	Visconti et al., 2005 [57]	Fe	Case–control	Italy	12/12	28 ± 8/28 ± 9	8/4; 7/5	Serum	1318 ± 527/1686 ± 547
34.	11.	Ay et al., 2023 [58]	Fe	Case–control	Türkiye	16/16	45 ± 7/44 ±12	9/7; 11/5	Serum	1762 ± 747/1226 ± 347
35.	12.	Gellein et al., 2008 [44]	Fe	Case–control	Norway	9/13	44 ± 3/49 ± 4	9/0; 8/5	Plasma	1058 ± 196/1053 ± 111
36.	13.	Al-Radaideh et al., 2021 [66]	Fe	Case–control	Jordan	65/34	18–58/20–60	41/24; 18/16	Serum	560 ± 40/833 ± 84
37.	14	Doğan and Yildiz, 2019 [67]	Fe	Case–control	Türkiye	53/45	37 ± 10/35 ± 10	11/42; 10/35	Serum	538 ± 110/794 ± 101
38.	15.	Abo-Krysha et al., 2008 [60]	Fe	Case–control	Egypt	20/10	29.94 ± 8.84	20/0; 10/0	Serum	595 ± 305/595 ± 466
39.	16.	Bergsland et al., 2017 [68]	Fe	Case–control	Italy	22/24	46.3/50.1	13/9; 17/7	Serum	1120 ± 660/1320 ± 740
40.	1.	Stojsavljević et al., 2024 [37]	Co	Case–control	Sebia	215/100	43 ± 10/42 ± 9	120/95; 66/34	Serum	0.574 ± 0.238/1.316 ± 0.630
41.	2.	Alimonti et al., 2007 [50]	Co	Case–control	Italy	60/124	39 ± 10/45 ± 13	38/22; 43/81	Serum	0.14 ± 0.03/0.l16 ± 0.03
42.	3.	Forte et al., 2005 [69]	Co	Case–control	Italy	60/60	39 ± 10/38 ± 10	40/20; N.A.	Plasma	0.22 ± 0.10/0.11 ± 0.06
43.	4.	Ristori et al., 2011 [56]	Co	Case–control	Italy	49/49	36 ± 7/33 ± 6	29/20; 26/23	Serum	0.17 ± 0.10/0.12 ± 0.12
44.	5.	Visconti et al., 2005 [57]	Co	Case–control	Italy	12/12	28 ± 8/28 ± 9	8/4; 7/5	Serum	0.21 ± 0.11/0.16 ± 0.09
45.	6.	Gellein et al., 2008 [44]	Co	Case–control	Norway	9/13	44 ± 3/49 ± 4	9/0; 8/5	Plasma	0.075 ± 0.014/0.066 ± 0.007
46.	1.	Stojsavljević et al., 2024 [37]	Cu	Case–control	Serbia	215/100	43 ± 10/42 ± 9	120/95; 66/34	Serum	873 ± 205/908 ± 131
47.	2.	Alimonti et al. 2007 [50]	Cu	Case–control	Italy	60/164	39 ± 10/45 ± 13	38/22; 43/81	Serum	938 ± 46/950 ± 51
48.	3.	De Riccardis et al., 2018 [70]	Cu	Case–control	Italy	38/39	N.A./N.A.	29/9; 18/21	Serum	1045 ± 45/820 ± 69
49.	4.	Forte et al., 2005 [69]	Cu	Case–control	Italy	60/60	39 ± 10/38 ± 10	40/20; N.A.	Plasma	1445 ± 481/926 ± 144
50.	5.	Ghoreishi et al., 2015 [49]	Cu	Case–control	Iran	50/50	32 ± 3/32 ± 3	N.A./N.A.	Serum	1882 ± 82/1031± 45
51.	6.	Kapaki et al., 1989 [41]	Cu	Case–control	Greece	15/28	34 ± 10/46 ± 14	3/12; 10/18	Serum	870 ± 150/1030 ± 140
52.	7.	Masoud and Fakharian, 2007 [43]	Cu	Cohort	Iran	35/35	32 ± 7/35 ± 5	28/7; 28/7	Serum	1160 ± 195/1337 ± 134
53.	8.	Siotto et al., 2019 [65]	Cu	Case–control	Italy	60/42	37 ± 9/40 ± 11	45/15; 22/20	Serum	900 ± 19/841 ± 22
54.	9.	Smith et al., 1989 [42]	Cu	Cohort	USA	27/33	N.A./N.A.	17/10; 11/22	Plasma	1736 ± 92/1735 ± 46
55.	10.	Socha et al., 2017 [55]	Cu	Case–control	Poland	101/68	41 ± 10/40 ± 13	64/37; 47/21	Serum	928 ± 398/988 ± 309
56.	11.	Ristori et al., 2011 [56]	Cu	Case–control	Italy	49/49	36 ± 7/33 ± 6	29/20; 26/23	Serum	916 ± 160/956 ± 193
57.	12.	Visconti et al., 2005 [57]	Cu	Case–control	Italy	12/12	28 ± 8/28 ± 9	8/4; 7/5	Serum	1034 ± 228/953 ± 75.2
58.	13.	Ay et al., 2023 [58]	Cu	Case–control	Türkiye	16/16	45 ± 7/44 ±12	9/7; 11/5	Serum	1179 ± 227/1268 ± 244
59.	14.	Nashmi et al., 2020 [59]	Cu	Case–control	Iraq	25/25	42 ± 9/41 ± 10	15/10; 14/11	Serum	1616 ± 159/1164 ± 195
60.	15.	Iranmanesh et al., 2012 [71]	Cu	Case–control	Iran	25/25	28 ± 3/N.A.	16/9; N.A.	Serum	886 ± 196/1104 ± 371
61.	16.	Palm and Hallmans, 1982 [38]	Cu	Case–control	Sweden	50/50	N.A./N.A.	29/21; 29/21	Serum	1000 ± 210/1071 ± 134
62.	17.	Ghazavi et al., 2012 [46]	Cu	Case–control	Iran	60/60	N.A./N.A.	43/17; 38/22	Serum	1152 ± 412/939 ± 296
63.	18.	Janghorbani et al., 2017 [72]	Cu	Case–control	Iran	55/95	32 ± 2/45 ± 2	47/8; 67/28	Plasma	1670 ± 223/1060 ± 195
64.	19.	Harbige et al., 2011 [45]	Cu	Case–control	UK	21/9	N.A./N.A.	N.A./N.A.	Plasma	1119 ± 309/957 ± 189
65.	20.	Gellein et al., 2008 [44]	Cu	Case–control	Norway	9/13	44 ± 3/49 ± 4	9/0; 8/5	Plasma	487 ± 117/478 ± 133
66.	21.	Rieder et al.,1983 [73]	Cu	Case–control	Switzerland	119/35	N.A./N.A.	63/53; 21/14	Plasma	888 ± 187/968 ± 198
67.	22.	Giacoppo et al., 2014 [48]	Cu	Case–control	Italy	41/23	41 ± 2/35 ± 2	31/10; 14/9	Plasma	623 ± 177/610 ± 199
68.	1.	Stojsavljević et al., 2024 [37]	Mn	Case–control	Serbia	215/100	43 ± 10/42 ± 9	120/95; 66/34	Serum	0.299 ± 0.284/4.390 ± 2.672
69.	2.	Alimonti et al., 2007 [50]	Mn	Case–control	Italy	60/124	39 ± 10/45 ± 13	38/22; 43/81	Serum	0.66 ± 0.08/0.60 ± 0.05
70.	3.	Forte et al., 2005 [69]	Mn	Case–control	Italy	60/60	39 ± 10/38 ± 10	40/20; N.A.	Plasma	8.39 ± 3.93/7.80 ± 3.00
71.	4.	Ristori et al., 2011 [56]	Mn	Case–control	Italy	49/49	36 ± 7/33 ± 6	29/20; 26/23	Serum	0.74 ± 0.30/0.57 ± 0.17
72.	5.	Visconti et al., 2005 [57]	Mn	Case–control	Italy	12/12	28 ± 8/28 ± 9	8/4; 7/5	Serum	1.13 ± 0.33/0.63 ± 0.16
73.	6.	Janghorbani et al., 2017 [72]	Mn	Case–control	Iran	55/95	32 ± 2/45 ± 2	47/8; 67/28	Plasma	0.94 ± 0.009/0.96 ± 0.007
74.	7.	Gellein et al., 2008 [44]	Mn	Case–control	Norway	9/13	44 ± 3/49 ± 4	9/0; 8/5	Plasma	2.64 ± 0.58/3.09 ± 0.35
75.	1.	Stojsavljević et al., 2024 [37]	Se	Case–control	Serbia	215/100	43 ± 10/42 ± 9	120/95; 66/34	Serum	72.230 ± 18.792/86.706 ± 11.684
76.	2.	Jensen et al., 1980 [74]	Se	Case–control	Denmark	14/12	N.A./N.A.	N.A./N.A.	Serum	85 ± 10/97 ± 14
77.	3.	Korpela et al., 1989 [75]	Se	Cohort	Finland	12/7	46 ± 9/N.A.	N.A./N.A.	Serum	123 ± 17/120 ± 18
78.	4.	Mazzella et al., 1983 [76]	Se	Case–control	Italy	20/16	37 ± 9/41 ± 6	14/6; N.A.	Plasma	86.4 ± 16.02/60.6 ±17.5
79.	5.	Smith et al., 1989 [42]	Se	Cohort	USA	27/33	N.A./N.A.	17/10; 11/22	Plasma	99 ± 3.95/104 ± 0.79
80.	6.	Socha et al.,2014 [77]	Se	Case–control	Poland	101/63	41 ± 10/41 ± 14	64/37; 43/20	Serum	55.2 ± 16.2/79.2 ± 20.6
81.	7.	Gellein et al., 2008 [44]	Se	Case–control	Norway	9/13	44 ± 3/49 ± 4	9/0; 8/5	Plasma	45.4 ± 2.1/41.1 ± 2.2
82.	8.	Giacoppo et al., 2014 [48]	Se	Case–control	Italy	41/23	41 ± 2/35 ± 2	31/10; 14/9	Plasma	68.60 ± 19.02/71.10 ± 18.09
83.	9.	Goli et al., 2023 [78]	Se	Case–control	Iran	70/70	N.A.	N.A.	Serum	85 ± 40/123 ± 60
84.	10.	Wilkström et al., 1976 [79]	Se	Case–control	Finland	27/18	N.A.	N.A.	Serum	46.4 ± 12.9/43.6 ± 13.0

Number of cases: Zn: 1308; Fe: 609; Co: 405; Cu: 1143; Mn: 460; Se: 536. Total number of cases: 4461. Number of controls: Zn: 1096; Fe: 595; Co: 358; Cu: 1031; Mn: 453; Se: 355. Total number of controls: 3888. Total number of participants for each element (cases + controls): Zn: 2404; Fe: 1204; Co: 763; Cu: 2174; Mn: 913; Se: 891. Total number of participants in the meta-analysis: 8349. N/A = not addressed.

**Table 2 biomedicines-12-01589-t002:** Quality assessment of studies included in the meta-analysis of Zn, Fe, Co, Cu, Mn, and Se levels in cases and controls: based on the Newcastle–Ottawa scale for case–control, cohort, and cross-sectional studies.

Study	Selection			Comparability		Outcome	Score	
	Representativeness	Size	Non-Respondents		Determination of Outcome	Statistical Test	For Biological Matrix	Average per Group
Zn								
Stojsavljević et al., 2024 [37]	a	a	a	a	a	a	6	
Palm and Hallmans, 1982 [38]	a	a	b	a	a	a	5	
Dore-Duffy et al., 1983 [39]	a	a	b	a	a	a	5	
Ho et al., 1986 [40]	a	a	c	a	a	a	4	
Kapaki et al., 1989 [41]	a	a	a	a	a	a	6	
Smith et al., 1989 [42]	b	a	b	a	a	a	4	
Masoud and Fakharian, 2007 [43]	b	a	a	a	a	a	5	
Gellein et al., 2008 [44]	a	a	a	a	a	a	6	
Harbige et al., 2011 [45]	a	a	c	a	a	a	4	
Ghazavi et al., 2012 [46]	a	a	b	a	a	a	5	
Al-Zubaidi, 2012 [47]	a	a	c	a	a	a	4	
Giacoppo et al., 2014 [48]	a	a	a	a	a	a	6	
Ghoreishi et al., 2015 [49]	a	a	b	a	a	a	5	
Alimonti et al., 2007 [50]	a	a	a	a	a	a	6	
Matar et al., 2020 [51]	a	a	a	a	a	a	6	
Mezzaroba et al., 2020 [52]	a	a	a	a	a	a	6	
Oraby et al., 2019 [53]	a	a	a	a	a	a	6	
Pawlitzki et al., 2018 [54]	b	a	a	a	a	a	5	
Socha et al., 2017 [55]	a	a	a	a	a	a	6	
Ristori et al., 2011 [56]	a	a	a	a	a	a	6	
Visconti et al., 2005 [57]	a	a	a	a	a	a	6	
Ay et al., 2023 [58]	a	a	a	a	a	a	6	
Nashmi et al., 2020 [59]	a	a	a	a	a	a	6	5.39
Fe								
Alimonti et al., 2007 [50]	a	a	a	a	a	a	6	
Abo-Krysha and Rashed, 2008 [60]	a	a	b	a	a	a	5	
Armon-Omer et al., 2019 [61]	b	a	a	a	a	a	5	
Bsteh et al., 2019 [62]	a	a	b	a	a	a	5	
Ellidag et al., 2014 [63]	a	a	a	a	a	a	6	
Matar et al., 2020 [51]	a	a	a	a	a	a	6	
Sfagos et al., 2005 [64]	a	a	c	a	a	a	4	
Siotto et al., 2019 [65]	a	a	a	a	a	a	6	
Ristori et al., 2011 [56]	a	a	a	a	a	a	6	
Visconti et al., 2005 [57]	a	a	a	a	a	a	6	
Ay et al., 2023 [58]	a	a	a	a	a	a	6	
Gellein et al., 2008 [44]	a	a	a	a	a	a	6	
Al-Radaideh et al., 2021 [66]	a	a	b	a	a	a	5	
Doğan and Yildiz, 2019 [67]	a	a	a	a	a	a	6	
Abo-Krysha et al., 2008 [60]	a	a	a	a	a	a	6	
Bergsland et al., 2017 [68]	a	a	a	a	a	a	6	5.62
Co								
Stojsavljević et al., 2024 [37]	a	a	a	a	a	a	6	
Alimonti et al., 2007 [50]	a	a	a	a	a	a	6	
Forte et al., 2005 [69]	a	a	b	a	a	a	5	
Ristori et al., 2011 [56]	a	a	a	a	a	a	6	
Visconti et al., 2005 [57]	a	a	a	a	a	a	6	
Gellein et al., 2008 [44]	a	a	a	a	a	a	6	5.83
Cu								
Stojsavljević et al., 2024 [37]	a	a	a	a	a	a	6	
Alimonti et al. 2007 [50]	a	a	a	a	a	a	6	
De Riccardis et al., 2018 [70]	a	a	b	a	a	a	5	
Forte et al., 2005 [69]	a	a	b	a	a	a	5	
Ghoreishi et al., 2015 [49]	a	a	b	a	a	a	5	
Kapaki et al., 1989 [41]	a	a	a	a	a	a	6	
Masoud and Fakharian, 2007 [43]	b	a	a	a	a	a	5	
Siotto et al., 2019 [65]	a	a	a	a	a	a	6	
Smith et al., 1989 [42]	b	a	b	a	a	a	4	
Socha et al., 2017 [55]	a	a	a	a	a	a	6	
Ristori et al., 2011 [56]	a	a	a	a	a	a	6	
Visconti et al., 2005 [57]	a	a	a	a	a	a	6	
Ay et al., 2023 [58]	a	a	a	a	a	a	6	
Nashmi et al., 2020 [59]	a	a	a	a	a	a	6	
Iranmanesh et al., 2012 [71]	a	a	c	a	a	a	4	
Palm and Hallmans, 1982 [38]	a	a	b	a	a	a	5	
Ghazavi et al., 2012 [46]	a	a	b	a	a	a	5	
Janghorbani et al., 2017	a	a	a	a	a	a	6	
Harbige et al., 2011 [45]	a	a	c	a	a	a	4	
Gellein et al., 2008 [44]	a	a	a	a	a	a	6	
Rieder et al.,1983 [73]	a	a	b	a	a	a	5	
Giacoppo et al., 2014 [48]	a	a	a	a	a	a	6	5.41
Mn								
Stojsavljević et al., 2024 [37]	a	a	a	a	a	a	6	
Alimonti et al., 2007 [50]	a	a	a	a	a	a	6	
Forte et al., 2005 [69]	a	a	b	a	a	a	5	
Ristori et al., 2011 [56]	a	a	a	a	a	a	6	
Visconti et al., 2005 [57]	a	a	a	a	a	a	6	
Janghorbani et al., 2017 [72]	a	a	a	a	a	a	6	
Gellein et al., 2008 [44]	a	a	a	a	a	a	6	5.86
Se								
Stojsavljević et al., 2024 [37]	a	a	a	a	a	a	6	
Jensen et al., 1980. [74]	a	a	c	a	a	a	4	
Korpela et al., 1989 [75]	b	a	c	a	a	a	3	
Mazzella et al., 1983 [76]	a	a	b	a	a	a	5	
Smith et al., 1989 [42]	b	a	b	a	a	a	4	
Socha et al.,2014 [77]	a	a	a	a	a	a	6	
Gellein et al., 2008 [44]	a	a	a	a	a	a	6	
Giacoppo et al., 2014 [48]	a	a	a	a	a	a	6	
Goli et al., 2023 [78]	a	a	c	a	a	a	4	
Wilkström et al., 1976 [79]	a	a	c	a	a	a	4	4.80
							Average of the study:	5.48

Selection: (1) Representativeness of the sample: a—truly representative of the average of the target population; b—reasonably representative of the average of the target population; c—selected group of users; d—no description of the sampling strategy. (2) Sample size: a—satisfactory; b—unsatisfactory. (3) Nonrespondents: a—comparability between the characteristics of the respondents and the nonrespondents is given and the response rate is satisfactory; b—the response rate is not satisfactory or the comparability between the respondents and the nonrespondents is not satisfactory; c—no description of the response rate or the characteristics of the respondents and nonrespondents. Comparability: (1) Comparability of subjects based on design or analysis: a—the study controls for the main factor; b—the study controls for each additional factor. Outcome: (1) Determination of outcome: a—independent blind assessment; b—record linkage; c—self-report; d—no description. (2) Statistical test: a—the statistical test used to analyze the data is clearly described and appropriate, and the measure of association is presented, including confidence intervals and probability level (*p*-value); b—the statistical test is inappropriate, not described, or incomplete. Calculation: a = 1; b—from the sum of sample type 1 is subtracted from the maximum number, 6; c—from the sum of sample type 2 is subtracted from the maximum number, 6. Modified quality assessment criteria according to Nakhaee et al. (2023) [80].

**Table 3 biomedicines-12-01589-t003:** The number of studies with statistically significant changes in Zn, Fe, Co, Cu, Mn, and Se levels in the serum of patients with MS (cases) and controls compared with the total number of studies included in the meta-analysis. Evidence of trace elements’ effects on cases based on meta-analysis results.

	Total Number of Analyzed Studies	Higher in Cases Than in Controls	Lower in Cases Than in Controls	No Statistically Significant Difference between Cases and Controls	Pooled Size Effects
Zn	23	3	10	10	Significantly lower; *p* = 0.012 *
Fe	16	2	7	7	Significantly lower; *p* = 0.007 *
Co	6	1	4	1	No significant changes; *p* = 0.326
Cu	22	8	4	10	Significantly higher; *p* = 0.002 *
Mn	7	3	3	1	No significant changes; *p* = 0.590
Se	10	2	4	2	No significant changes; *p* = 0.351

* Significantly different at *p* < 0.05.

## Data Availability

The original contributions presented in the study are included in the article, further inquiries can be directed to the corresponding author.

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
