# Peer review of "Changes of Target Essential Trace Elements in Multiple Sclerosis: A Systematic Review and Meta-Analysis"

_biomedicines, 2024, doi:10.3390/biomedicines12071589_

Round 1
Reviewer 1 Report
Comments and Suggestions for Authors
The idea is interesting, however, the manuscript should be written more clear. There are some examples that may help:
-The review question (aim) of the study is not clearly mentioned in introduction.
-In the abstract, the aim of this study is not mentioned at all.
-What do authors mean by the association of trace elements with MS? What are the effect sizes that have been searched and pooled? Is that risk ratio? odds ration? SMD?
-How did the authors concluded these highly-heterogene results may be considered in supplementation of pwMS?
-The results in the abstract should be reported along with statistical numbers.
-The search syntax is incorrect. Why there is AND between all components? Should a single study report all of these components together? What about a study which only assessed zinc in MS?
- The results section should be summarized in tables. No need to report all findings in the manuscript.
-There should be more info from MS group in the table. i.e, their MS type, disease duration, EDSS. How these factor were considered in the analysis?
- What do you mean by 'abnormal data' in PRISMA?
Author Response
The idea is interesting, however, the manuscript should be written more clear. There are some examples that may help:
- The review question (aim) of the study is not clearly mentioned in introduction.
Answer: The aim has been reformulated. Thank you.
- In the abstract, the aim of this study is not mentioned at all.
Answer: The aim has been reformulated. Many thanks.
- What do authors mean by the association of trace elements with MS?
Answer: Thank you. The sentence has been deleted.
- What are the effect sizes that have been searched and pooled? Is that risk ratio? odds ration? SMD?
Answer: Effect size is a statistical concept that measures the strength of the relationship between two variables on a numerical scale, as the studies included in a meta-analysis must have common outcome statistics for their results to be combined. The larger the effect size, the greater the differences. The effect size in meta-analysis refers to different studies and then combines all studies into a single analysis. If all studies to be included in a meta-analysis have the same outcome measure, an effect size can be calculated in the original units. If the effect sizes are available in the original units, the interpretation is clearer. The overall effect size derived from the meta-analysis is calculated by combining the effect sizes of the included studies. There are several types of effect sizes: Pearson’s r correlation, standardized mean difference, Cohen’s d, Hedges’ g method, Cramer’s φ. Odds ratios or relative risks are also used for effect sizes. In addition, the effect size of individual studies may be somewhat imprecise and therefore lead to an unstable result if several small studies are used. Finally, variables such as gender, age differences or differences in the intervention administered, e.g. dose, may influence the size and direction of the effect size (https://www.jospt.org/doi/10.2519/jospt.2011.3333). In our meta-analysis, based on other similar meta-analyzes, we decided to use Hedges’ g method as a measure of effect size.
- How did the authors concluded these highly-heterogene results may be considered in supplementation of pwMS?
Answer: Many meta-analyzes have a similar problem: pooling and analyzing the results of researchers from around the world who have used different sample sizes, different analysis techniques, gender structures, age groups, etc., which affects the heterogeneity of the results. For this reason, the data obtained through a meta-analysis in many cases show a great deal of heterogeneity. However, when we use Hedges' g and the pooled effect size, we can find significant differences between groups despite this heterogeneity. Therefore, heterogeneity is not the only factor determining further recommendations as we proposed for pwMS.
Heterogeneity is a term used to describe variability among studies. There are statistical and clinical heterogeneity. We estimate statistical heterogeneity that occurs when the treatment effect estimates of a set of studies vary among one another. Clinical heterogeneity refers to differences in study methods that affect the ability to compare and/or combine data from different studies. Examples of differences in study methods that may lead to clinical heterogeneity include differences in participant demographics, such as risk or severity of disease, the settings in which the research was conducted, the frequency and intensity of the intervention, and how outcomes were measured across studies. While there are statistical tests to estimate the extent of statistical heterogeneity, there are no tests to determine the extent of clinical heterogeneity.
In some instances, authors may decide not to conduct a meta-analysis because the heterogeneity is too great. Other authors may attempt to minimize heterogeneity within a meta-analysis by limiting study eligibility. One of the methods is to decide which studies should be included in the meta-analysis before starting the study, based on acceptable statistical and clinical heterogeneity and the quality of assessment for inclusion. This approach, while reducing heterogeneity, typically results in the total number of articles included on a topic to be reduced. In our meta-analysis, we decided to minimize heterogeneity by reducing the number of studies included in the analysis.
-The results in the abstract should be reported along with statistical numbers.
Answer: Thank you. Recent practice has shown that it is not important to burden the abstract with excess text. In addition, the PRISMA protocol we followed does not require presentation of results in the abstract.
-The search syntax is incorrect. Why there is AND between all components? Should a single study report all of these components together? What about a study which only assessed zinc in MS?
Answer: This is the strategy defined by the PRIZMA protocol that we have used. Thus, the word “AND” refers not only to the two words it connects, but generally between all key words. On the other hand, some studies examined only one trace element, some several, etc. Thank you.
- The results section should be summarized in tables. No need to report all findings in the manuscript.
Answer: When reviewing a large number of meta-analyzes, the textual explanation of the results obtained predominates. Of course, reading such results is rather dry and tedious, and a tabular presentation is more transparent. However, even the PRISMA protocol that we followed does not require the results to be presented in tabular form, so we decided to present them according to the rules that apply to meta-analyzes. Thank you.
-There should be more info from MS group in the table. i.e, their MS type, disease duration, EDSS. How these factor were considered in the analysis?
Answer: Thank you. Not all studies provided the sufficient data mentioned by the reviewer (MS type, disease duration, EDSS, etc.) so we decided not to include them in the meta-analysis, as this would leave us with a very small amount of available literature data for the analysis itself and many results would not be valid in this case. Our main goal was to collect a sufficient number of papers with the concentrations of the studied trace elements independent of the MS types. This leaves us the possibility that, if there are enough such papers, we may also perform subgroups within the meta-analysis.
- What do you mean by 'abnormal data' in PRISMA?
Answer: By abnormal data the PRIZMA meant extremely low or extremely high concentrations of trace elements in certain studies (of which there were only a few, namely only 3). Many thanks for taking the time to review our manuscript.
Reviewer 2 Report
Comments and Suggestions for Authors
Stojsavljević et al. presents an important study in examining changes in essential trace elements that are associated with the pathogenesis of multiple sclerosis. The study is conducted with comprehensive meta-analysis which provides insights into the difference in the trace elements between disease and controls. The study is impactful and broad audience. I recommend publication after addressing some minor comments that might help to improve the manuscript.
(1) It might be important to discuss the nature of the studies. For example, are the studies discussed making use of omics analysis (e.g., metabolomics) or cellular/molecular neuroscience approaches.
(2) Similar to the comments above, it would be good to discuss how the use of metabolomics or similar high dimensional techniques or data mining of these studies may further provide validations for this study. Some references that might be useful: (a) https://www.sciencedirect.com/science/article/pii/S2095177923001235; (b) https://www.mdpi.com/1422-0067/22/20/11112; (c) https://www.mdpi.com/2076-3425/13/9/1318
(3) While the authors mentioned that some trace elements are significantly changed in MS, are there any interactions between these elements? For example, do some of these elements change simultaneously or have some correlation between one another? This may provide more information with regards to the disease mechanism.
(4) It would be good to include the clinical courses and progression of each study in Table 1. For example, RRMS, SPMS, or PPMS.
(5) It would be good to further elaborate on the use of trace elements for biomarker discovery for various stages of MS.
(6) The study highlights personalized and targeted strategies in the management of MS which is very important. However, it is also important to highlight the heterogeneity of disease phenotypes and course of progression in MS patients and hence the need for precision or personalized medicine. Some references that might be useful: (a) https://www.neurology.org/doi/full/10.1212/NXI.0000000000200025; (b) https://doi.org/10.1177/1352458519881558; (c) https://www.frontiersin.org/articles/10.3389/fncel.2021.726479/full
Comments on the Quality of English Language
N/A
Author Response
Comments and Suggestions for Authors
Stojsavljević et al. presents an important study in examining changes in essential trace elements that are associated with the pathogenesis of multiple sclerosis. The study is conducted with comprehensive meta-analysis which provides insights into the difference in the trace elements between disease and controls. The study is impactful and broad audience. I recommend publication after addressing some minor comments that might help to improve the manuscript.
(1) It might be important to discuss the nature of the studies. For example, are the studies discussed making use of omics analysis (e.g., metabolomics) or cellular/molecular neuroscience approaches.
(2) Similar to the comments above, it would be good to discuss how the use of metabolomics or similar high dimensional techniques or data mining of these studies may further provide validations for this study. Some references that might be useful: (a) https://www.sciencedirect.com/science/article/pii/S2095177923001235; (b) https://www.mdpi.com/1422-0067/22/20/11112; (c) https://www.mdpi.com/2076-3425/13/9/1318
Answer: Thanks for both questions. However, metabolomics is not the goal of our research and the authors of this study have no expertise in that area. Therefore, we decided not to go to an area that is not close to us.
(3) While the authors mentioned that some trace elements are significantly changed in MS, are there any interactions between these elements? For example, do some of these elements change simultaneously or have some correlation between one another? This may provide more information with regards to the disease mechanism.
Answer: The main aim of the meta-analysis was to summarize the results obtained by other authors in a standardized way. Changes occur in the sense of an increase or decrease in the parameters monitored in the meta-analysis, and our task was to see which trend prevails with the help of special statistical procedures. In the discussion, it is possible to comment on our opinion of the problem, but not on the use of correlation or other analyzes. Thank you.
(4) It would be good to include the clinical courses and progression of each study in Table 1. For example, RRMS, SPMS, or PPMS.
Answer: Based on the available literature and the objectives set for this meta-analysis, we were not able to collect a sufficient number of papers that would provide us with valid data on RRMS, SPMS and PPMS that would be standardized enough to avoid excessive heterogeneity and bias in the analysis. For this reason, we started from the point of view that it is necessary to start a meta-analysis with the most available data. While this would be too general, it would certainly encourage other researchers to conduct analyzes of the studied trace elements on a larger scale on specific types of MS. Thank you.
(5) It would be good to further elaborate on the use of trace elements for biomarker discovery for various stages of MS.
Answer: Thank you. As we have already mentioned, this is a first step in the research of general differences between patients and healthy individuals, a kind of a pilot meta-analysis, which, as the reviewers themselves can tell and which is normal in science, raises many more new questions and answers only one question asked. In our case, we have obtained an answer to the differences between the trace elements between healthy individuals and MS patients, but other questions are also raised, such as: what are the differences between RRMS, SPMS and PPMS, what is the distribution of patients according to their place of residence and therefore the geological background that influences the concentration of the elements, and many others. We hope that we all find appropriate answers to these questions in the near future.
(6) The study highlights personalized and targeted strategies in the management of MS which is very important. However, it is also important to highlight the heterogeneity of disease phenotypes and course of progression in MS patients and hence the need for precision or personalized medicine. Some references that might be useful: (a) https://www.neurology.org/doi/full/10.1212/NXI.0000000000200025; (b) https://doi.org/10.1177/1352458519881558; (c) https://www.frontiersin.org/articles/10.3389/fncel.2021.726479/full
Answer: Many thanks to the reviewer for a very interesting observation. The heterogeneity of the disease has been a major problem for clinicians for years to find the right guidelines for the diagnosis and treatment of this disease. Surely, the only possible solution is to develop a personalized approach. The literature suggested by the reviewer is very useful for us in this sense, but, as indicated above, it was not possible to systematize all factors in one paper. All in all, many thanks for taking the time to review our manuscript.
Reviewer 3 Report
Comments and Suggestions for Authors
GENERAL COMMENT
The submitted review manuscript is comprehensive and well written.
METHODS
The methods and results are clearly presented
DISCUSSION
The authors suggest monitoring of trace elements in MS patients and supplementation in cases with low iron and zinc levels. This suggestion should be followed up with evidence of clinical benefit of such supplementation.
· Are there publications of the effect of lower zinc or iron on clinical outcome in MS? For example PMID: 30049983; PMID: 29396631
· Are there publications of the effect of supplementation of zinc or iron on clinical outcome in MS? For example PMID: 30963028
Author Response
Comments and Suggestions for Authors
GENERAL COMMENT
The submitted review manuscript is comprehensive and well written.
METHODS
The methods and results are clearly presented
DISCUSSION
The authors suggest monitoring of trace elements in MS patients and supplementation in cases with low iron and zinc levels. This suggestion should be followed up with evidence of clinical benefit of such supplementation.
- Are there publications of the effect of lower zinc or iron on clinical outcome in MS? For example PMID: 30049983; PMID: 29396631
- Are there publications of the effect of supplementation of zinc or iron on clinical outcome in MS? For example PMID: 30963028
Answer: We thank the reviewer for the important comment. Individual studies provide contradictory results, which is why we came up with the idea of carrying out a meta-analysis. For each element, there are papers in which some authors have identified increased or decreased concentrations of certain elements, but the final results indicate which data prevail. The situation is complicated due to the very heterogeneous clinical symptoms of the disease, the different areas where the patients live and therefore the different availability of elements in the geological background and in the diet, as well as the differences in the recommended doses for supplementation. All this shows how little we know about this disease and that it is caused by complex mechanisms that indicate not only a possible outcome of supplementation, but a combination of complex and intertwined biochemical mechanisms that are possible in the development and treatment of the disease. Also, perhaps most importantly, studies dealing with supplementation were not the target of our meta-analysis. We believe that future studies will address this issue. Overall, many thanks for the kind words and questions.
Round 2
Reviewer 1 Report
Comments and Suggestions for Authors
Unfortunately, my comments have not been addressed well. Thank you.
Author Response
-